# Creativity or Brute Force?
# Using 🧠Brainteasers as a Window into the
# Problem-Solving Abilities of Large Language Models

**Simeng Han**[1♡]    **Howard Dai**[1♡]    **Stephen Xia**[1♡]    **Grant Zhang**[1♡]

**Chen Liu**[1♡]    **Lichang Chen**[2]    **Hoang Huy Nguyen**[3]

**Hongyuan Mei**[4]    **Jiayuan Mao**[5]    **R. Thomas McCoy**[1]

[1]Yale University    [2]Meta Superintelligence Labs
[3] Georgia Institute of Technology    [4]TTIC    [5]Massachusetts Institute of Technology

[♡]Lead contributors.
Please direct correspondence to `sophiahsm6@gmail.com`.

## Abstract

Accuracy remains a standard metric for evaluating AI systems, but it offers limited insight into how models arrive at their solutions. In this work, we introduce a benchmark based on brainteasers written in long narrative form to probe more deeply into the types of reasoning strategies that models employ. Brainteasers are well-suited for this goal because they can be solved with multiple approaches, such as a few-step solution that uses a creative insight or a longer solution that uses more brute force. We investigate large language models (LLMs) across multiple layers of reasoning, focusing not only on correctness but also on the quality and creativity of their solutions. We investigate many aspects of the reasoning process: (1) semantic parsing of the brainteasers into precise mathematical competition-style formats; (2) self-correcting solutions based on ground-truth solutions; (3) producing step-by-step sketches of solutions; and (4) making use of hints. We find that LLMs are in many cases able to find creative, insightful solutions to brainteasers, suggesting that they capture some of the capacities needed to solve novel problems in creative ways. Nonetheless, there also remain situations where they rely on brute force, despite the availability of more efficient, creative solutions, highlighting a potential direction for improving LLM reasoning.

## 1 Introduction

Recent techniques for improving large language models (LLMs) have highlighted an important fact: In order to improve AI systems, we should think carefully about *how* they arrive at their solutions. For instance, many of the most popular recent enhancements to LLMs are approaches for guiding the models to solve problems in certain ways, such as decomposing tasks hierarchically into simpler sub-tasks, leveraging an internal monologue (commonly known as a chain of thought) to perform self-reflection and verify generated solutions [1, 2, 3], or translating problems from a verbal description into formal representations before attempting solutions [4, 5]. One reason why the solution strategy is so important is that some strategies may be more generalizable than others. For instance, brute-force strategies might generalize less well to complex, novel problems than the ability to form creative insights. Therefore, understanding how a model arrives at its solutions is helpful for understanding the generality of the model's abilities.

---

The code is available at `https://github.com/stephenxia1/brainteasers`.
The dataset is available at `https://huggingface.co/datasets/ChenLiu1996/Brainteaser`.

39th Conference on Neural Information Processing Systems (NeurIPS 2025).

> **Problem: Six Village (from the BRAINGLE BRAINTEASER Math subset)**
>
> There are six villages along the coast of the only perfectly round island in the known universe. The villages are evenly distributed along the coastline so that the distance between any two neighboring coastal villages is always the same. There is an absolutely straight path through the jungle connecting every pair of villages. These paths create thirteen crossings in the interior of the island, one of which is in the middle of the island where paths from every village meet. The island has a strange courtship custom. Before a father will give permission for his daughter to marry, her suitor must bring the father a fish each day until he has traveled by every route from his village to the father's village. The young man only travels along routes where he is always getting closer to his destination. The young man may visit other villages along the way.
>
> On April 1st a father's three sons come to tell him of their intent to woo a bride, each from a different village. The brides' villages are the first three villages encountered when traveling clockwise around the island. If the sons begin their courtship today and the couples are married on the day following each son's last trip, what are the three wedding dates?

Despite the importance of solution strategy, there has been little work on evaluating the types of heuristics and solution components that LLMs use. Most LLM evaluations use benchmarks based on accuracy [6, 7, 8], which provide important insight into how well LLMs perform on particular tasks but do not illuminate how they approach those tasks. A few prior papers have taken a finer-grained look at the reasoning process by evaluating intermediate steps to more faithfully measure reasoning capabilities [9, 10, 11].

The contrast between brute-force and insightful strategies is evident in where the "intelligence" is applied. A brute-force or exhaustive approach leverages the model's speed to try many possibilities, which can be useful for combinatorial problems or when the solution can be verified easily. An insightful approach requires the model to recognize a clever trick, pattern, or underlying principle that leads directly to the answer, mimicking human insight. In current LLMs, we often see a mixture of both. For instance, an LLM might attempt a quick enumeration of options if a direct reasoning trick doesn't immediately present itself.

In this work, we perform a systematic analysis of LLM reasoning strategies through a novel benchmark dataset that we introduce, BRAINGLE BRAINTEASER, which uses brainteasers to evaluate the reasoning abilities of LLMs. The types of brainteasers that we use are effective for this purpose because solving them involves multiple aspects of reasoning (e.g., translating language into mathematical abstractions and finding creative insights) and because they can be solved in multiple different ways (e.g., creative, insight-based approaches or longer brute-force approaches). Therefore, these brainteasers are well-suited for illuminating the types of reasoning processes that LLMs tend to use and the aspects of reasoning that they perform well on or struggle with. BRAINGLE BRAINTEASER is focused on evaluating mathematical and logical reasoning with minimal knowledge barriers, distinguishing it from existing datasets like MATH [7] and AIME [12] by emphasizing sequential problem-solving over knowledge recall. Authored by expert problem solvers, BRAINGLE BRAINTEASER features diverse puzzle styles and complexities, aiming to isolate models' reasoning abilities rather than their memorization of formulas. BRAINGLE BRAINTEASER is exclusively centered on mathematical and logical puzzles, all authored by expert problem solvers with demonstrated proficiency across a wide range of puzzle types. Consequently, the benchmark exhibits greater diversity in both problem style and complexity than previous logical puzzle benchmarks [13, 14, 15]. Using this benchmark, we systematically evaluate LLM reasoning performance by decomposing reasoning into distinct sub-tasks: translating narrative (verbal) descriptions into mathematical descriptions, generating solutions directly from both verbal and mathematical representations, verifying solutions against ground truths, generating high-level solution plans, summarizing detailed solutions into conceptual outlines, and finally, generating solutions given such high-level plans.

Our key contributions are twofold. First, we present BRAINGLE BRAINTEASER, a structured dataset that allows for systematic probing of many aspects of the reasoning pipeline. Second, our empirical analysis with OpenAI o3 [16], DeepSeek R1, DeepSeek R1 Distilled models [17], DeepSeek V3 [18], and Gemini Flash [19] provides clear evidence regarding heuristics that are commonly used in reasoning research. We find that: 1) LLMs sometimes default to brute force in cases where more

creative solutions are available. This could be because the training objective is aimed at optimizing the final answer accuracy instead of optimizing for generating more creative solutions or the correctness of the reasoning trace [17]. 2) LLMs struggle to correct solutions based on official solutions when prompted to do so. 3) Translating from verbal narratives into mathematical-style problem statements provides modest gains in performance. 4) The strongest reasoning models are able to reliably break down the solutions into insightful steps and models are capable of using high-level steps to generate correct solutions for the hardest problems where they fail to generate solutions fully from scratch.

Overall, our findings challenge widely held heuristic assumptions, such as the presumed simplicity of task decomposition, verification, and formal translation. Therefore, future research on reasoning models should evaluate these assumptions and shift from heuristic-driven algorithms towards systematic compositions and improvements of LLM capabilities.

## 2    Related Works

**LLM Reasoning Benchmarks**    Existing mathematical reasoning benchmarks span a wide spectrum of difficulty, from grade-school arithmetic [7, 8] to Math Olympiad-level problems [20, 21, 22] and collegiate contests (e.g., the Putnam competition) [23]. [24] provides a formal-proof benchmark at the Olympiad level and the AlphaGeometry suite focuses specifically on IMO-level geometry challenges [25, 26]. In contrast, our benchmark, BRAINGLE BRAINTEASER, is challenging (like prior benchmarks focusing on Olympiad problems or collegiate contests) but without requiring the substantial domain knowledge that is demanded in prior challenging benchmarks. It achieves its difficulty by emphasizing the generation of creative insights, isolating an LLM's capacity for novel solution generation rather than its access to sophisticated mathematical facts. By posing concrete problems, rather than the sometimes highly abstract statements typical of Olympiads and collegiate contests, BRAINGLE BRAINTEASER makes it easier both for models to verify long or complex answer strings and for evaluators to assess correctness. Distinct from recently introduced human-written or synthetic puzzle benchmarks [13, 14, 15], BRAINGLE BRAINTEASER is specifically focused on mathematical and logical puzzles, and written by human experts proficient in solving diverse types of puzzles. They are also diverse in problem style as well as problem complexity. Finally, our results show that BRAINGLE BRAINTEASER is non-saturated, which suggests that it is not overly prone to data contamination [27, 28], creating the opportunity to identify both strengths and weaknesses of LLM reasoning.

**Reasoning Evaluation Beyond Final Answer Accuracy**    Recent research has underscored the need to evaluate an LLM's reasoning process itself—rather than focusing solely on final-answer accuracy—by examining intermediate steps to more faithfully measure reasoning capabilities [9, 11, 29, 30]. In this work, rather than directly evaluating intermediate steps, we systematically explore four facets that are closely related to the capabilities of LLM reasoning such as self-correction of generated solutions based on official solutions and strategic use of hints to guide problem solving.

**Reasoning Creativity**    Previous studies have explored enhancing LLM creativity in solving mathematical conjectures by leveraging programmatic approaches and formal self-refinement within symbolic domains [31, 32, 33]. In contrast, our work evaluates the creativity manifested in LLM-generated natural language explanations, assessing whether they introduce novel insights that effectively guide and simplify the problem-solving process.

## 3    The BRAINGLE BRAINTEASER Benchmark

In this section we provide qualitative and quantitative descriptions of the BRAINGLE BRAINTEASER benchmark, which is made of two sub-datasets: a Math dataset and a Logic dataset.

### 3.1    Dataset Construction

The Math and Logic datasets were curated by scraping problem-solving and reasoning questions from the Braingle[1] website, an online platform of puzzles and brain teasers. The problems on Braingle are largely curated by human experts proficient in solving diverse types of logical puzzles. We have obtained permission from the website owners to distribute the specific examples used in this study. To systematically extract the data, we implemented a web scraper that crawls through all available puzzle pages and recorded the question-answer pairs. We specifically select the math and logic subsets in

---

[1]https://www.braingle.com/brainteasers/All.html

Table 1: Puzzle counts by category for Math and Logic datasets. Popularity and Difficulty are both on a scale from 1 (least popular/difficult) to 4 (most popular/difficult).

(a) Puzzle count by category in the Math set.

| Category | Count | Popularity | Difficulty |
|---|---|---|---|
| **Standard** | | | |
| Geometry | 24 | 2.19 ± 0.18 | 2.87 ± 0.19 |
| Number Theory | 34 | 2.35 ± 0.22 | 2.91 ± 0.15 |
| Combinatorics | 24 | 2.27 ± 0.11 | 2.76 ± 0.10 |
| Algebra | 56 | 2.37 ± 0.22 | 2.75 ± 0.12 |
| **Nonstandard** | | | |
| Logic | 29 | 2.35 ± 0.23 | 2.77 ± 0.14 |
| Special Number | 29 | 2.35 ± 0.22 | 2.78 ± 0.13 |
| **Heuristic** | | | |
| Pattern | 28 | 2.28 ± 0.22 | 2.82 ± 0.15 |
| Arithmetic | 18 | 2.44 ± 0.24 | 2.80 ± 0.19 |
| **Total** | 242 | 2.33 ± 0.22 | 2.80 ± 0.15 |

(b) Puzzle count by category in the Logic set.

| Category | Count | Popularity | Difficulty |
|---|---|---|---|
| **Simple/large** | | | |
| 0D | 29 | 2.53 ± 0.23 | 2.68 ± 0.23 |
| 1D | 13 | 2.56 ± 0.30 | 2.60 ± 0.28 |
| 2D | 22 | 2.47 ± 0.30 | 2.72 ± 0.25 |
| Number | 17 | 2.43 ± 0.27 | 2.74 ± 0.26 |
| Clusters | 8 | 2.56 ± 0.20 | 2.78 ± 0.13 |
| Tree | 6 | 2.83 ± 0.17 | 2.69 ± 0.15 |
| **Complex/small** | | | |
| Liars | 17 | 2.50 ± 0.28 | 2.66 ± 0.22 |
| Communication | 4 | 2.70 ± 0.28 | 2.55 ± 0.15 |
| Compound | 9 | 2.48 ± 0.44 | 2.67 ± 0.34 |
| **Math-like** | | | |
| Algorithm | 38 | 2.56 ± 0.34 | 2.66 ± 0.23 |
| Math | 32 | 2.41 ± 0.27 | 2.63 ± 0.22 |
| **Heuristic** | | | |
| Pattern | 26 | 2.48 ± 0.27 | 2.63 ± 0.22 |
| Linguistic | 15 | 2.50 ± 0.12 | 2.61 ± 0.15 |
| **Total** | 236 | 2.51 ± 0.28 | 2.66 ± 0.23 |

Braingle which are designed as logical and mathematical reasoning challenges. We also collected associated metadata such as titles, user-voted difficulty ratings, popularity scores, and optional hints, creating a rich and structured dataset for analysis. In our datasets, we select the top 250 most difficult problems in the math and logic categories for our Math and Logic datasets respectively, as these represent the most challenging problems in each category.

As quality control, we conducted one round of manual inspection of all problems in BRAINGLE BRAINTEASER done by college students who belong to a math club; these students have extensive experience in solving competition-level math problems. During manual inspection, low-quality ambiguously-described problems were removed, leaving 242 Math and 236 Logic problems in the dataset. The same annotators also manually created hints for problems that originally lacked them.

## 3.2 Detailed Dataset Information

**Population Statistics** Table 1 provides statistics of BRAINGLE BRAINTEASER. The most difficult problem has a user-voted difficulty score of 3.06. All problems have a difficulty level above 2.30 as we only consider the most challenging problems in our dataset. The average number of words in human solutions is greater in Logic (172 words) than in Math (237 words).

**Problem Categorization** We divide the Math top 250 dataset into 3 categories and 8 subcategories shown in Table 1a. In general, the **Standard** category describes problems which can be written as standard math test or competitive math problem forms albeit requiring less knowledge. The **Nonstandard** category describes problems which still have a rigorous answer, but do not fit the form of a standard competitive math problem. The **Heuristic** category describes problems which are not mathematically rigorous, but require creative pattern-finding or heuristic thinking. We divide the Logic dataset into 4 categories and 13 subcategories shown in Table 1b. In general, the **Simple/large** category describes large deductive logic puzzles, which often contain long chains of reasoning. The **Complex/small** category also describes deductive logic puzzles, often with fewer possible answers, but with highly complex clues. The **Math-like** category describes puzzles which do not have fixed/bounded state spaces, where problems require mathematical calculation or algorithmic design. The **Heuristic** category mirrors that of the heuristic category in the math set: informal problems which require creative pattern-finding. We also note that human-provided difficulty ratings are relatively consistent across categories. Full analysis and explanation for both Math and Logic categorizations can be found in Appendices H, I, respectively.

Table 2: Mean and st. dev. of solution step counts for Braingle Math and Logic datasets.

| | Solution Steps | Median | Creative Steps | Rudimentary Steps |
|---|---|---|---|---|
| Braingle Math | 6.4 ± 2.5 | 6.0 | 2.0 ± 1.2 | 4.4 ± 2.2 |
| Braingle Logic | 8.6 ± 5.1 | 7.0 | 2.4 ± 1.7 | 6.2 ± 4.7 |

**Distribution of Number of Steps** We also provide the distribution of number of steps in human solutions as part of our benchmark. We can approximate our understanding of the complexity of brainteasers and the required creativity to solve them by counting the number of steps in the brainteasers' provided human solutions. We define a step to be a key component of the problem,

but this does not necessarily imply that every individual deduction is a step. We observe that, by providing OpenAI o3 with detailed instructions and few-shot examples of step breakdown, it is able to restate the human solutions into correct, broken-down steps that arrive at the final answer based on manual inspection on 30 examples. This is also true for problems that o3 fails to solve correctly on its own.

We also make the distinction between creative and rudimentary steps. A creative step generates innovative insights that reduce the problem or make the problem significantly easier to solve. Common traits of creative steps include using analogies, combining ideas from different domains, exploiting problem constraints, or devising elegant, efficient strategies that go beyond straightforward computation or trial-and-error. A rudimentary step often is more easily or routinely derivable, such as straightforward computation; it could also be a step that may make progress in solving the problem but is not innovative, such as using trial-and-error or systematically exploring all possible options. We provide the prompt with few-shot examples in Appendix G. Table 2 provides statistics regarding the distribution of step count for all the problems in BRAINGLE BRAINTEASER.

### 3.3 Problem Example

We provide a representative example of a problem from BRAINGLE BRAINTEASER. Unlike traditional math and reasoning datasets, BRAINGLE BRAINTEASER problems may be informal, open-ended, and pattern-based. For example, consider the following problem from the Math dataset:

---

**Problem: Math 4**

What characteristic do these three 12-digit numbers share with each other, but with no other 12-digit number?

$$100307124369$$
$$111824028801$$
$$433800063225$$

---

Any arbitrary "characteristic" may be defined, so there is no rigorous mathematical solution. However, there is a clear answer up to human reasonability, which is to observe the following pattern:

---

**Human Solution: Math 4 (abbreviated)**

The sum of their digits are square numbers:

$$1 + 0 + 0 + 3 + 0 + 7 + 1 + 2 + 4 + 3 + 6 + 9 = 36 = 6^2$$
$$1 + 1 + 1 + 8 + 2 + 4 + 0 + 2 + 8 + 8 + 0 + 1 = 36 = 6^2$$
$$4 + 3 + 3 + 8 + 0 + 0 + 0 + 6 + 3 + 2 + 2 + 5 = 36 = 6^2$$

The sum of their digit pairs are square numbers:

$$10 + 03 + 07 + 12 + 43 + 69 = 144 = 12^2$$
$$11 + 18 + 24 + 02 + 88 + 01 = 144 = 12^2$$
$$43 + 38 + 00 + 06 + 32 + 25 = 144 = 12^2$$

Similarly, the sum of their digit triplets, quadruplets, and sextuplets, as well as the numbers themselves, are all square numbers.

---

The inclusion of these informal and heuristically-driven problems in BRAINGLE BRAINTEASER allows us to assess LLMs not only on their ability to construct formal solutions to structured math and logic problems, but also on their ability to find creative patterns which align with human intuition.

## 4 Experiments and Analyses

We evaluate a range of LLMs on BRAINGLE BRAINTEASER. We first discuss model accuracy on the benchmark and then present more detailed, finer-grained analyses of the reasoning process.

Table 3: Model solution correctness, in percent, under different prompting strategies on the math and logic datasets. Results for the entire subset are shown, whereas the results for top 50 most difficult problems are displayed in parentheses.

| Dataset | Model | CoT Prompt | Math Prompt | w Hint | Math Prompt w Hint |
|---------|-------|-----------|-------------|--------|--------------------|
| Math | DeepSeek R1 Distill Qwen 1.5B | 17.2 (14.0) | 16.4 (10.0) | 15.2 (8.0) | 17.6 (10.0) |
| | DeepSeek R1 Distill Qwen 14B | 41.2 (22.0) | 44.0 (30.0) | 44.0 (20.0) | 42.6 (26.0) |
| | DeepSeek R1 Distill Llama 70B | 42.4 (20.0) | 40.8 (22.0) | 45.6 (24.0) | 44.2 (18.0) |
| | deepseek-chat (Deepseek-V3) | 58.0 (46.0) | 55.6 (38.0) | 56.0 (36.0) | 58.8 (36.0) |
| | deepseek-reasoner (Deepseek-R1) | 66.8 (48.0) | 70.2 (54.0) | 72.4 (48.0) | 72.8 (56.0) |
| | gemini-2.5-flash-preview-04-17 | 66.0 (34.0) | 65.2 (38.0) | 69.2 (44.0) | 72.0 (58.0) |
| | OpenAI o3 | 79.6 (66.0) | 79.6 (64.0) | 82.8 (66.0) | 81.2 (68.0) |
| Logic | DeepSeek R1 Distill Qwen 1.5B | 4.0 (4.0) | 4.0 (6.0) | 6.8 (6.0) | 3.6 (4.1) |
| | DeepSeek R1 Distill Qwen 14B | 22.0 (16.0) | 23.6 (16.0) | 27.2 (22.0) | 26.0 (32.0) |
| | DeepSeek R1 Distill Llama 70B | 24.4 (16.0) | 24.4 (14.0) | 26.0 (20.0) | 29.2 (26.0) |
| | deepseek-chat (Deepseek-V3) | 37.8 (30.6) | 40.8 (28.0) | 41.6 (22.0) | 41.4 (24.5) |
| | deepseek-reasoner (Deepseek-R1) | 44.6 (26.0) | 45.4 (32) | 49.4 (32.7) | 50.6 (40.0) |
| | gemini-2.5-flash-preview-04-17 | 49.2 (36.0) | 51.2 (34.0) | 54.0 (42.0) | 53.6 (38.0) |
| | OpenAI o3 | 68.4 (50.0) | 71.2 (54.0) | 70.0 (52.0) | 74.4 (54.0) |

## 4.1 Solution Correctness

**Evaluation Approach** Instead of manually grading model responses, we assessed whether OpenAI o3 [16] could correctly evaluate model-generated solutions by comparing them to human solutions. We carefully verified that o3's judgments aligned with human labels with 99.3% and 97% average raw agreement for solution correctness and creativity and brute-force distinction respectively. Therefore, we adopt the model solution accuracy assessed by o3 as our primary correctness metric. Details on LLM-human agreement experiments are in Appendix B.1.

**Prompting Approaches** We evaluated models under several prompting approaches. The CoT Prompt encourages the model to generate a step-by-step solution [34]. The Math Prompt additionally encourages the model to use rigorous mathematical reasoning, explicitly discouraging brute force, guesswork, and shortcuts. It emphasizes the need for step-by-step logical justification, ensuring each inference aligns strictly with the problem conditions. For each of these prompts, we included a version with a hint or without a hint, where the hint provides a problem-specific hint that either came from Braingle or our annotators (see Section 3.1). The prompt texts can be found in Appendix F.

**Model Performance** The results (Table 3) show that all models that we evaluated display a non-trivial degree of success on this dataset, but in all cases there remains meaningful room for improvement. Larger models such as OpenAI o3, deepseek-reasoner, and gemini-2.5-flash-preview-04-17 consistently outperform smaller models like DeepSeek R1 Distill Qwen 1.5B, especially on the Math dataset. However, despite these improvements, even the most capable models still struggle to achieve high accuracy on the most difficult subset of problems (top-50 accuracy shown in parentheses). For example, OpenAI o3, while achieving 88.8% on the broader Math set with hints and math-specific prompting, only reaches 80.5% on the top-50 problems, reflecting the persistent difficulty of these harder questions. OpenAI o3 does not return responses for some of the hardest problems even when given an unbounded amount of time to think; this could be because that the problem is too hard for the model to answer.

Another key trend is that providing hints usually boosts model performance across all sizes and datasets. This effect is particularly evident when comparing the 'w Hint' and 'Math Prompt w Hint' conditions to their non-hint counterparts. In both Math and Logic tasks, hints help models achieve better reasoning steps and overcome some of their inherent limitations in tackling complex reasoning without guidance. There is a performance gap between 'CoT Prompt' and 'w hint' even in some of the strongest models, indicating that some insights that are necessary for solving the problems remain elusive such that models benefit from hints that impart these insights.

We present details on model performance by category in Appendix J.

## 4.2 Brute-force vs. Creativity

Humans often resort to brute-force when they cannot immediately solve a problem, as validating a guessed solution is typically easier than deriving one from first principles. Compared to humans, models have access to more significant computational resources, allowing them to guess-and-check

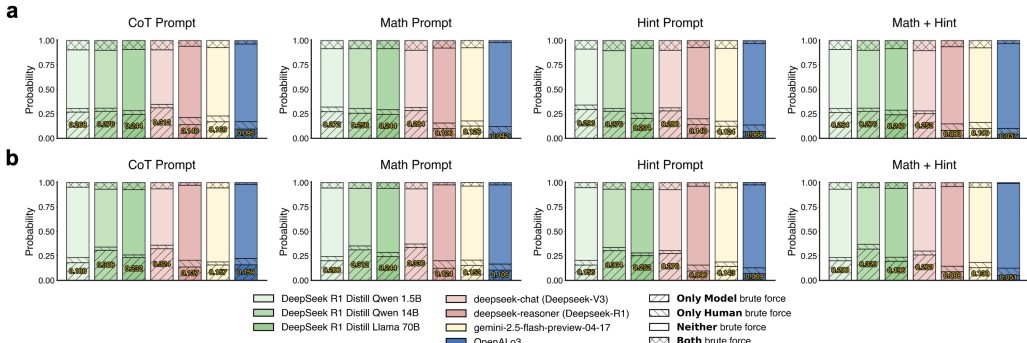

Figure 1: Distributions of brute-force and non-brute-force solutions. Results are shown for the **(a) Math** and the **(b) Logic** datasets. Numbers reflect cases where only the model used brute force.

efficiently. However, this strategy reflects a less sophisticated form of reasoning. When possible, it would be ideal for a model to instead identify key insights, such as insights that reduce the problem into manageable sub-components, because the ability to form such insights may be necessary in novel, complex situations where brute force is impossible or intractable. In this section, we analyze the extent to which the models that we evaluated rely on brute force. For clarity, below we define how we distinguish between brute force and creativity.

**Brute-force Solution**    We define a brute-force solution as a simple, exhaustive search strategy that systematically explores possibilities until an answer is found. This approach is viable when the search space is small but does not scale to more complex problems. Typical characteristics of brute-force solutions include reliance on code, guess-and-check procedures, or computations beyond what a human could feasibly do. For example, when asked to find the smallest positive integer satisfying a given condition, OpenAI `o3` occasionally resorts to simulating the problem environment and iteratively testing values until a solution is found. While effective in reaching an answer, such guess-and-check approaches typically lack rigorous solutions and adequate reasoning.

**Creative Solution**    We define a creative solution as an innovative, insight-driven approach that leverages pattern recognition or lateral thinking. Rather than exhaustively testing all possibilities, it reframes the problem or exploits shortcuts to reduce complexity. Such solutions often involve minimal computation and are especially valuable for problems where brute-force search is intractable.

**Evaluation**    We evaluated model reasoning strategies by prompting OpenAI `o3` to judge whether each solution uses brute force. We used a few-shot prompt that includes labeled examples of brute-force and non-brute-force reasoning to guide classification. We assessed reasoning behaviors across two tasks (`Math` and `Logic`), four prompt types (`CoT`, `Math`, `Hint`, `Math+Hint`), and seven models. The complete results are shown in Figure 1, with breakdowns in Appendix Tables S6 and S7.

All models use non-brute-force strategies on a sizable proportion of the problems. Nonetheless, on all prompts, models also sometimes resort to brute force, including on problems where the corresponding human solutions from Braingle do not. However, the tendency to use brute force decreases with model size and capability increasing. For example, OpenAI `o3` and DeepSeek Reasoner show much lower brute force rates than smaller models such as DeepSeek Qwen 1.5B or Gemini 2.5 Flash. This suggests larger models are better able to avoid guess-and-check procedures and apply more structured reasoning. Still, brute force behavior appears across all models, indicating this is a common fallback even for advanced systems. See Appendix K.3 for a breakdown of which problems models are most likely to use brute-force on.

Prompting has a strong influence on brute-force behavior. The `Math` prompt consistently reduces brute force usage compared to the `CoT` baseline, suggesting that framing a problem in mathematical terms encourages more deliberate reasoning. However, the effect is limited. In both datasets, adding a `Hint` also reduces reliance on brute force. The combined prompt `Math+Hint` yields the greatest reduction in brute force reliance across all models, indicating that combining structured framing with helpful cues is especially effective.

These results highlight two levers for improving model reasoning in the direction of becoming more creative and less reliant on brute force: scaling and prompting. While larger models appear naturally

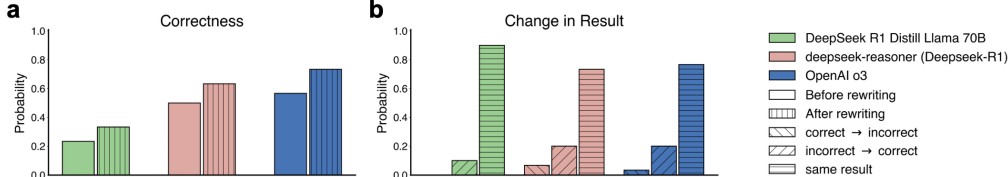

Figure 2: Effect of rewriting on the top 30 most difficult rewritable math brainteasers in their original narrative format ("before rewriting") and then rewritten into a more mathematical format ("after rewriting"). **(a)** Rewriting slightly increases model correctness. **(b)** For all models, most results remain the same after rewriting, while a small fraction of initially incorrect answers are corrected.

less reliant on brute-force, prompting remains a powerful mechanism for aligning model behavior with human-like problem-solving expectations.

**Brute-force vs Adequate/Inadequate Summary**   When models resort to brute force, do they do so arbitrarily or because the more creative solution lies outside the model's abilities? To answer this question, we analyze whether models can summarize the solutions to the problems (see Table S12). The percentage of problems where models use brute-force is higher when the model is unable to accurately summarize a solution, which suggests that the same reasoning barrier that incentivizes a model to choose brute-force also causes it to be unable to understand the human solution. This supports the hypothesis that these models still have some limitations in creative problem-solving.

### 4.3   Informed self-correction based on the human solution

We further analyzed the "informed self-correction" capabilities of LLMs. Specifically, we investigate (1) whether LLMs are able to identify errors and propose corrections by comparing a flawed LLM solution ($S_{\text{LLM}}$) with the correct human solution ($S_{\text{Human}}$), and (2) whether LLMs will be fooled if we swap the flawed and correct solutions in this process (prompt shown below). We prompted all models to perform informed self-correction, and manually inspected the top 50 most difficult examples, which yielded 14 examples where $S_{\text{LLM}}$ is incorrect. For fair comparison, in all experiments, we define $S_{\text{LLM}}$ as the OpenAI o3 solution with CoT Prompt.

**Error Analysis**   When correcting a flawed $S_{\text{LLM}}$ using $S_{\text{Human}}$ (Figure S5a), all models were able to admit that $S_{\text{LLM}}$ was wrong (low fault denial). However, DeepSeek R1 distillation models, especially the 1.5B variant, were prone to degenerative repetition. Error misattribution also occurred: models often failed to locate the exact error. Overall, 3/6 models produced valid corrections in over 80% of cases. Such valid corrections were more common in models with a lower brute-force rate in Section 4.2. In particular, OpenAI o3, deepseek-chat, and deepseek-reasoner are more likely to detect the error, propose an appropriate fix, and present a complete and coherent solution. When tricked to "correct" $S_{\text{Human}}$ using $S_{\text{LLM}}$, all models took the bait and accepted the premise in most cases (Figure S5b). Over 60% of the time, models were **overly agreeable** and were reluctant to challenge the prompt, and therefore almost always unquestioningly, but incorrectly, accepted that $S_{\text{Human}}$ was incorrect (false confession). Justified denial (correctly rejecting the flawed $S_{\text{LLM}}$ and insisting $S_{\text{Human}}$ was correct) was very rare. Case studies of these error modes are in Appendix L.

### 4.4   Translating from verbal narratives into mathematical-style problem statements

Brainteasers typically are narratives instead of concise, symbolic problems common in math competitions. To see whether stylistic difference affects model performance, we rewrite problems in the Math subset, reformulating them into a competition-style format without altering the core logic.

**Rewritability**   We first manually categorize each problem as rewritable or non-rewritable. Problems are deemed non-rewritable if they (1) lack a rigorous mathematical answer (e.g., involve open-ended pattern discovery), (2) already resemble standard competition problems, or (3) are as concise as possible in their current form. From the set of rewritable problems, we selected the 30 most challenging ones and prompted OpenAI o3 to rewrite each into a symbolic, competition-style version. Manual inspection confirmed that 24 of these rewriting attempts were mathematically equivalent to the original version, while the remaining 6 contained only minor omissions (e.g., missing a constraint),

Table 4: Mean & standard deviation of solution step counts for the 30 hardest Braingle Math puzzles.

| | Solution Steps | Median | Creative Steps | Rudimentary Steps |
|---|---|---|---|---|
| OpenAI o3 Solution ($n = 29$)* | $7.7 \pm 2.9$ | 8.0 | $2.0 \pm 1.0$ | $5.7 \pm 3.0$ |
| OpenAI o3 Correct Solution($n = 21$) | $8.0 \pm 2.2$ | 8.0 | $1.9 \pm 0.9$ | $6.1 \pm 2.4$ |
| DeepSeek** V3 Solution ($n = 26$)* | $12.4 \pm 5.4$ | 12.0 | $4.0 \pm 3.3$ | $8.5 \pm 6.1$ |
| DeepSeek V3 Correct Solution ($n = 10$) | $10.9 \pm 2.4$ | 10.5 | $3.3 \pm 1.5$ | $7.6 \pm 2.2$ |
| Llama 70B Solution ($n = 21$)* | $10.7 \pm 3.4$ | 10.0 | $2.9 \pm 1.7$ | $7.8 \pm 3.7$ |
| Llama 70B Correct Solution ($n = 6$) | $8.7 \pm 2.9$ | 9.5 | $2.0 \pm 0.6$ | $6.7 \pm 2.8$ |

suggesting a strong underlying semantic grasp. We corrected these 6 cases that had minor omissions, resulting in fully-valid rewritings for all 30 problems.

**Evaluation**   We evaluated performance on the rewritten problems using OpenAI o3, deepseek-reasoner, and DeepSeek-R1-Distill-LLaMA-70B, with the same CoT prompt (Fig. 2). This isolates the effect of narrative style from mathematical complexity. Rewriting problems into a more formal mathematical format increased correctness across all models. o3 improved from 56.7% to 73.3% ($p = 0.023$), and deepSeek-reasoner from 50.0% to 63.3% ($p = 0.043$), both statistically significant. DeepSeek-R1-Distill-LLaMA-70B showed a smaller, non-significant gain (23.3% to 33.3%, $p = 0.08$). This suggests LLMs better interpret formal statements when they are expressed mathematically instead of narratively. Thus, interpreting formal statements that are presented narratively remains an area of improvement for LLMs, and for now this difficulty can be ameliorated by encouraging the LLM to first rewrite problem statements in a mathematical format. However, some problems were already near-formal or inherently complex, limiting overall improvement.

### 4.5   Automatic breakdown of solutions

We next analyzed the number of steps that the models used to arrive at their solutions. We quantified the number of steps using OpenAI o3. Before doing so, we manually evaluated whether OpenAI o3 was able to break down the correct human-written solutions into more granular steps that arrive at the final answer (Appendix G). We find that OpenAI o3 is able to break the solutions down in this way, counting steps at the level of granularity that we desired, in which each step is a meaningful insight rather than counting every new statement as a step. For example, for math problems that require choosing between several candidates to deduce mystery numbers, the models consider the analysis of all candidates as one collective step rather than dedicating one step for analyzing each candidate. For logic puzzles with a given list of clues, the use of a clue is considered a single step, even when multiple strings of thought are made (which is as intended). The example steps are listed in Appendix G. We find that OpenAI o3 is successful at counting steps in the desired way even for problems that o3 fails to solve correctly independently.

Table 4 compares the mean and standard deviation of OpenAI o3's step breakdown of OpenAI o3, DeepSeek V3 and Llama 70B model solutions[2]. We can see that DeepSeek solutions have the most steps and that all models' solutions contain considerably more rudimentary than creative steps. While DeepSeek R1 and Llama 70B have generally low accuracy, o3 is still able to correctly divide the model outputs into steps while maintaining the same logic regardless of correctness. Furthermore, OpenAI o3 is able to divide the models' solutions responses into steps that more closely represent key components of solving the problems rather than individual deductions.

## 5   Representative Case Study and Solution Error Patterns

In this section, we complement the systematic, quantitative analyses presented above with some qualitative analyses of the specific types of errors that models exhibit.

### 5.1   Inefficient Use of Hints

A key qualitative gap between human and machine solutions lies in how hints are prioritized when multiple hints are provided. OpenAI o3 often defers the most informative hint until the very end, forcing itself to sift through an unnecessarily large search space. We provide an example of such

---

[2]Several model solutions were removed either for being outliers due to excessive casework, or for failing to provide a complete solution that OpenAI o3 could break down; DeepSeek V3 was used over DeepSeek R3 as DeepSeek R3 would too often only present the final answer rather than a full solution.

Table 5: Case study for o3's insufficient usage of hints in the scenario where a model is unable to form the correct reasoning chain.

| Hint | o3 solution excerpt |
|---|---|
| "Take the sum of the number, the number rotated one to the left, the number rotated one to the right, the number with the first three and last three digits swapped, the number with the digit pairs rotated to the left, and the number with the digit pairs rotated to the right. The first and last digits of this sum match the last two digits of the number, in some order." | 416723, 167234 (rotate L), 341672 (rotate R), 723416 (swap), 672341 (rotate pairs L), 234167 (rotate pairs R). The sum is 2555553, first digit is 2, last digit is 3, exactly the two digits e=2 and f=3 of the original number. The condition is satisfied. |

phenomena with a six-digit number-guessing puzzle. One of the provided hints for this puzzle is shown in "Hint" in Table 5.

This hint is long but encodes a simple reduction. Note that if the sum of digits is some 2-digit $XY$, the sum is equivalent to $111111 \cdot XY$. The last digit of the sum and the last digit of the original number are thus guaranteed to match, and the first digit of the sum matches the second-to-last digit if and only if no carryover occurs in this product, which occurs if $X + Y < 10$. This hint is by far the most informative, as two other hints in the problem relate digits to the value of the digit sum: 1) "The digital sum matches the number formed by the last two digits in the number." 2) "The sum of the first two digits is the same as the sum of the last two digits."

This is the hint first considered in the provided human solution, and it results in a very quick reduction to a small search pool. However, in the o3 solution, this hint is used as the final filter, leading to a much larger list of candidates considered. This is likely due to the length of the hint itself; in general, it is most effective to check more complex hints at the end of a problem after more immediate deductions are made, so it may be a natural heuristic to use the textually longest hints last; however, in this case, a helpful hint has been wrapped in a convoluted description. Even during the check at the end, the model does not recognize the helpful reduction and opts to write out each cycle and calculate by hand, as shown in "o3 solution excerpt" in Table 5.

**Explanation of Behavior** The model appears to rank hints by textual length or syntactic complexity — saving the longest, "hardest-looking" clues for last. When a concise mathematical pivot is wrapped in wordy prose, o3 consequently undervalues it. More effective prompting should encourage the model to evaluate hints based on informational content, not surface length, or explicitly instruct it to test all hints for potential early eliminations.

### 5.2 Recurring Error Patterns in OpenAI o3 Solutions and Prompt-based Remedies

OpenAI o3 frequently mirrors the "hand-waving" shortcuts that humans sometimes slip into proofs—skipping essential justifications and filling the gaps with confident but hollow phrases. We identify three prominent patterns, and present prompt instructions that mitigate each, steering the model toward fuller justifications and greater rigor. However, these prompt instructions do not necessarily lead to correct solutions. We provide detailed examples of such errors and our prompting strategies in Appendix M.

## 6 Conclusion

We have introduced BRAINGLE BRAINTEASER, a novel benchmark of expert-authored mathematical and logical puzzles designed to probe distinct facets of LLM reasoning, including semantic parsing, solution generation, correction, step breakdown, and hint utilization. Through comprehensive experiments with OpenAI o3, DeepSeek variants and Gemini Flash, we uncover several key insights. First, despite opportunities for creative shortcuts, LLMs frequently default to exhaustive search, especially on harder problems, indicating a reliance on computational power rather than creative insight. Second, structured hints yield consistent accuracy gains—particularly for strong models— by guiding chain-of-thought toward critical intermediate observations, whereas simple brute-force discouragement prompts yield minimal benefit. Third, models can identify and correct their own errors when provided with official solutions, but remain susceptible to "false confession" when flawed outputs are presented as ground truth, underscoring challenges in robust self-evaluation. Fourth, converting narrative puzzles into strict mathematical formulas does not reliably boost downstream solution accuracy, suggesting that semantic translation alone is insufficient to enhance reasoning. Finally, advanced reasoning models are capable of breaking down solutions reliably and generating correct solutions given a step summary for the most challenging problems.

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

# Supplementary Materials for
# Creativity or Brute Force?
# Using 🧠Brainteasers as a Window into the
# Problem-Solving Abilities of Large Language Models

# A Limitations

Our study's scope has several limitations. The effectiveness of prompt engineering was constrained, as simple discouragement of brute-force had minimal impact, and even hints showed varied success. Our findings are specific to the BRAINGLE BRAINTEASER benchmark, derived mainly from one source and focused on math/logic puzzles, which may limit generalizability. The use of OpenAI o3 for certain evaluations and our macro-level analysis of solution step counts could introduce biases or mask detailed computational efforts.

Beyond these, LLMs frequently defaulted to brute-force over creative insight, faced challenges in robust self-correction (including "false confessions"), and didn't consistently improve when puzzles were formalized mathematically. These observations, from a specific set of LLMs and our operational definitions of creativity and brute-force, highlight current boundaries in advanced LLM reasoning.

In addition, because our dataset was taken from the public Braingle website, data contamination is a potential concern. Initial pilot studies showed that, on the few hardest problems in both Logic and Math datasets, frontier models such as OpenAI o3 often constructed solutions (either correct or incorrect) which were highly distinct from the human solutions on the website. While this is a potential indicator against high levels of data contamination, this is entirely anecdotal and further study should be conducted on how our results may be affected by data contamination.

# B LLM-as-a-Judge

We recognize that our use of OpenAI o3 as a judge may create potential biases in our evaluation process. We first note that using LLM-as-a-Judge is an established practice, with an extensive number of papers showing that LLMs, especially recent reasoning models have high accuracy at such judgments on reasoning tasks [35, 36, 37, 38, 39, 40, 41]. Combined with the fact that o3 aligned well with our human annotators, we believe that this choice is well-justified.

## B.1 LLM-Human Agreement Experiments

To address these concerns, we conducted a human evaluation with three annotators on 100 o3-generated solutions. Human annotators annotate each solution for correctness and whether it reflected creative or brute-force reasoning. We first provide a description of the human annotators:

1) The annotators are college students who are native English speakers. 2) They have formal education in mathematical and logical reasoning, majoring in Mathematics or Statistics. 3) They have all undergone more than five years of training in solving competition-level math problems, each receiving individual top 500 honors in the proof-based Putnam competition. In addition, they are all current or past organizing board members of a university math competition, and thus have extensive experience in problem writing with a focus on designing problems with creative and elegant (non-brute-force) solutions.

We provided the annotators with the same detailed annotation guidelines as those given to o3, including clear definitions and examples of brute-force and creative solutions.

We find that o3's judgments aligned with human labels with 99.3% and 97% average raw agreement for solution correctness and creativity/brute-force distinction respectively, reinforcing its reliability as an automatic evaluator in this setting. We also find that o3's judgment has higher alignment than Claude-3.7, deepseek-reasoner, Gemini 2.5 Flash and GPT-4o.

# C Broader Impact

This work provides a systematic framework for probing the reasoning capabilities of large language models (LLMs) through the lens of brainteasers, offering insights into how these systems approach complex problem-solving tasks. By distinguishing between creative and brute-force strategies, the BRAINTEASER benchmark encourages the development of models that reason more efficiently and human-like, rather than relying on computational brute force. This shift toward more nuanced reasoning abilities could enhance AI's applicability in fields requiring critical thinking, such as education, scientific discovery, and collaborative problem-solving, where transparency and interpretability of reasoning processes are essential.

# D  Full Dataset Statistics

Table S1: Population statistics of the datasets curated.

| Statistics↓  Data→ | Braingle Math Most difficult ($n = 250$) | Braingle Logic Most difficult ($n = 250$) |
|---|---|---|
| Percentage with hints | 100% | 100% |
| Difficulty score out of 4 | $2.80 \pm 0.15$ | $2.66 \pm 0.23$ |
| median | 2.79 | 2.64 |
| Answer word count | $172 \pm 204$ | $237 \pm 236$ |
| median | 109.5 | 162.5 |
| Answer sentence count | $7.01 \pm 8.13$ | $10.88 \pm 10.91$ |
| median | 4.00 | 8.00 |

# E  API Inference Settings

| Model | $\text{Top}_p$ | Max Tokens | Temperature |
|---|---|---|---|
| DeepSeek R1 Distill Qwen 1.5B | 0.7 | 10000 | 0.7 |
| DeepSeek R1 Distill Qwen 14B | 0.7 | 10000 | 0.7 |
| DeepSeek R1 Distill Llama 70B | 0.7 | 10000 | 0.7 |
| deepseek-chat (Deepseek-V3) | 1 | 10000 | 1 |
| deepseek-reasoner (Deepseek-R1) | - | 10000 | 1 |
| gemini-2.5-flash-preview-04-17 | 0.95 | 10000 | 1 |
| OpenAI o3 | 1 | 10000 | 1 |

# F  Prompts

---

**Prompt: Chain of Thought (CoT)**

You are an excellent mathematician with perfect logic. You are also very patient, and willing to perform very long chains of reasoning when necessary.
Can you solve this problem? Please spell out your entire reasoning steps. Finish your response in the format of "Final answer: " immediately followed by your answer.

---

**Prompt: Math Prompt**

You are an excellent mathematician with perfect logic. You are also very patient, and willing to perform very long chains of reasoning when necessary. Solve the given problem, keeping in mind the following:
If you use a brute force or guess-and-check method or utilize code when not necessary, you will receive no credit. If you do not fully justify a step, you will receive no credit.
Ensure each of your statements is consistent with the conditions of the problem and statements you have already written before moving on. If you do not do this, you will receive no credit.
An outline of a solution without a concrete final answer will also receive no credit.
The problem statement is correct, and a correct answer exists. If you solve a version of the problem or attempt to modify the problem statement to a version other than the one written exactly as-is, you will receive no credit.
Finish your response in the format of "Final answer: " immediately followed by your answer.

# G   Counting Steps

To have OpenAI o3 and DeepSeek R1 interpret solutions, break them down into steps, and label steps as creative or rudimentary, we presented the following prompt.

> **Prompt: Counting Steps**
>
> I will provide you with a problem and its solution. Without solving the problem yourself, I want you to divide the provided solution into steps. Then, I want you to distinguish each step as either a creative step or a rudimentary step. A creative step is a step that generates innovative insights that reduce the problem or make the problem significantly easier to solve. Common traits of creative steps include using analogies, combining ideas from different domains, exploiting problem constraints, or devising elegant, efficient strategies that go beyond straightforward computation or trial-and-error. A rudimentary step is a step that applies creative insights and often is more easily or routinely derivable, such as straightforward computation; it could also be a step that may make progress in solving the problem but is not innovative, such as using trial-and-error or systematically exploring all possible options. Common traits of rudimentary steps include utilizing code, guess-and-check, or performing computations that a human would ordinarily not be able to do. In terms of defining a step, note that a step is defined as something that does not need to be super fine-grained like "1 + 1 = 2" or "a -> b, b->c; a->c." Instead, each step should represent a key component of the solution and the steps sequentially lead to the final answer to form a complete solution. Make sure to not omit any necessary information, and also make sure that your standard of what is considered a "step" is as consistent as possible. Count the total number of steps in the human solution and report each step and the total number of steps in the format "Total Step Count:" followed by the number of steps. Also, keep count of the number of creative steps and rudimentary steps. After displaying the total step count , please report "Creative Steps:" followed by the number of creative steps and "Rudimentary Steps:" followed by the number of rudimentary steps. Lastly, report the steps themselves in the format "Steps:" with the steps needed to solve the problem.

As mentioned in section 3.2, models are often able to correctly comprehend the idea that a step being a key component of the problem does not necessarily imply that every individual deduction is a step. We now present explicit examples of steps that consist of multiple deductions which combine to a more key component of the problem. For example, for math problems that require choosing between several candidates to deduce mystery numbers, the models consider the analysis of all candidates as one collective step rather than dedicating one step for analyzing each candidate. Take the following output from OpenAI o3.

> **Model Response: Math 9 (Output Step Example)**
>
> For each of those products, ask whether Prashant could become certain after hearing Sachin's opening remark. Only product 52 qualifies, because every other product can be factored in more than one way that keeps the sum inside or outside the Possible-Sum list.

Note that multiple computations would be required in analyzing each individual candidate, let alone all candidates; even so, the model is able to combine these rudimentary, brute force-like calculations as an overall step, in turn adhering to our intended definition of a step of representing a substantial amount of progress to solving the problem rather than every individual deduction.

For logic puzzles with a given list of clues, the use of a clue is considered a single step, even when multiple strings of thought are made, which is, again, as intended.

---

**Model Response: Logic 5 (Output Step Example)**

Use the "only one ascending triple" condition (clue 7).
• Exactly 1 lower than Q can be left of her (or 2 triples would appear), triple must be Q–?–A.
• To prevent J-K-A or T-K-A from creating a second triple, K must be to the far left.
• To avoid T-J-A making another triple, J must precede T.
Complete rank order: K J Q T A.

---

Now, we consider the following problem that wants to arrange two 1's, 3's, 7's, and 9's in a string such that each pair of consecutive integers is prime, as well as the number itself and its reverse. We will then compare the outputs of OpenAI o3 and DeepSeek R1 to observe any similarities and differences in how solutions are divided into steps and how steps are categorized as either creative or rudimentary.

---

**Problem: Math 23**

Arrange the numerals "11337799" to form an 8-digit number (referred to as "ABCDEFGH") that meets the following conditions:
1) ABCDEFGH is a prime number 2) HGFEDCBA (reversing the digits) is a prime number 3) all consecutive 2-digit subsets of ABCDEFGH are prime numbers (AB, BC, CD, DE, EF, FG, GH are all prime) 4) all consecutive 2-digit subsets of HGFEDCBA are prime numbers (HG, GF, FE, ED, DC, CB, BA are all prime)

---

Note that the creative step uses a mathematical insight regarding reversible primes containing 9 to reduce the problem into more limited space of possibilities, while the rudimentary step simply uses trial-and-error on all possibilities of the remaining digits to find valid combinations. Again, the explicit steps themselves are provided below.

---

**Human Solution: Math 23**

The only available 2-digit prime numbers are: 11, 13, 17, 19, 31, 37, 71, 73, 79, and 97. The number pairs that are prime in both directions are: 11/11, 13/31, 17/71, 37/73, and 79/97 (19 is eliminated since its reverse, 91, is not prime). With these numbers, it can be seen that 9 can only be paired with 7. Therefore, the solution must be of the form 97????79, 9797????, or ????7979. ???? must therefore be 1313, 3131, or 3113. This leaves the possible solution pairs of 97131379/97313179, 97971313/31317979, 97973131/13137979, 97973113/31137979, and 97311379/97311379 (the last pair being a palindrome). Of these, only 97131379 and its reverse, 97313179, are prime numbers.

---

**Model Response: Math 23 (OpenAI o3)**

1. List every two–digit prime that can be built from the available digits 1, 3, 7, 9: 11, 13, 17, 19, 31, 37, 71, 73, 79, 97. (Rudimentary)

2. From that list, keep only those whose reversals are also prime, because each two–digit block must work in both the forward and the reverse 8-digit number. The viable pairs are 11/11, 13/31, 17/71, 37/73, 79/97. (Rudimentary)

3. Observe that the digit 9 can appear in a two-digit prime only as 79 or 97, so every 9 must be adjacent to a 7. (Creative)

4. Use the previous observation to pin down the overall skeleton of the 8-digit number: it must be of the form 97 _ _ _ _ 79, 9797 _ _ _ _, or _ _ _ _ 7979. (Creative)

5. Insert the remaining digits 1,1,3,3 into the four open slots so that every new two-digit block is still in the list from Step 2, leaving only three feasible fillings: 1313, 3131, or 3113. (Rudimentary)

6. Combine the frameworks from Step 4 with the fillings from Step 5 to generate the complete candidate pairs of 8-digit numbers (each with its reverse): 97131379 / 97313179, 97971313 / 31317979, 97973131 / 13137979, 97973113 / 31137979, 97311379 / 97311379. (Rudimentary)

7. Test each candidate for primality; only 97131379 and its reverse 97313179 are prime, so they satisfy all conditions. (Rudimentary)

---

**Model Response: Math 23 (DeepSeek R1)**

1. List all possible 2-digit primes from the given digits (11, 13, 17, 19, 31, 37, 71, 73, 79, 97). (Rudimentary: This is a straightforward enumeration of primes from the given digits.)

2. Eliminate primes whose reverses are not prime (e.g., 19 is eliminated because 91 is not prime). (Creative: This insight reduces the problem by filtering out invalid candidates early.)

3. Identify digit constraints (e.g., 9 can only be paired with 7). (Creative: This observation narrows down the possible starting and ending digits of the solution.)

4. Determine possible forms of the solution (97????79, 9797????, or ????7979). (Creative: This step leverages the digit constraints to structure the problem into manageable subproblems.)

5. Enumerate possible middle segments (1313, 3131, or 3113). (Rudimentary: This is a systematic exploration of possible combinations.)

6. Generate all candidate pairs based on the forms and middle segments. (Rudimentary: This is a straightforward generation of possible numbers from the constraints.)

7. Check which candidates and their reverses are prime numbers. (Rudimentary: This involves computational verification of primality.)

8. Conclude that only 97131379 and its reverse 97313179 satisfy all conditions. (Creative: This final step synthesizes the results and confirms the unique solution.)

---

Note that both models divide the steps nearly identically, with the only difference being that the DeepSeek model divides the testing of final candidates and the declaration of the final answer as two steps whereas OpenAI o3 only considers this as one step. More importantly, both models' nearly identical breakdowns also correctly follow the human solution's logic in the exact sequence it was presented; this exemplifies the models' abilities to properly parse a provided response into steps while maintaining its overall logical structure. Now, we observe both models' steps 3, 4, and 5. Both models declare steps 3 and 4 as creative in making more sophisticated deductions on prime numbers including 9 and significantly narrowing down the possible forms of the solution, which correspond to our definition of creative steps in making an innovative insight without systematically exploring all possible options. Contrarily, both models correctly declare the brute-force exploration of feasible segments of 1's and 3's as a rudimentary step.

However, as noted in Table S2 and S3, we observe that DeepSeek R1 tends to declare more steps as creative than OpenAI o3, and we observe that here, DeepSeek declares steps 2 and 8 as creative while OpenAI o3 considers them to be rudimentary. By our definitions, we expect this to be more rudimentary; step 2 requires scanning all of the prime numbers to see which primes have prime reversals, which would be rudimentary, but DeepSeek considered this creative for filtering invalid candidates. Despite the significant progress, the greater focus should have been placed on the

Table S2: Mean and standard deviation of solution step counts for several models' (OpenAI o3, DeepSeek V1, Llama 70B) solutions for the most difficult puzzles in the Braingle Math and Braingle Logic datasets (n=250) based on OpenAI o3. DeepSeek V1 solutions were used instead of DeepSeek R1 solutions because too many DeepSeek R1 responses only consisted of the final answer without the reasoning process that we can divide into steps.

| | Solution Steps | Median | Creative Steps | Rudimentary Steps |
|---|---|---|---|---|
| **Braingle Math** | | | | |
| OpenAI o3 Solution | $8.2 \pm 3.5$ | 8.0 | $1.9 \pm 1.2$ | $6.3 \pm 3.2$ |
| DeepSeek V1 Solution | $11.2 \pm 4.6$ | 10.0 | $2.9 \pm 2.5$ | $8.3 \pm 3.7$ |
| Llama 70B Solution | $9.6 \pm 3.4$ | 9.0 | $2.6 \pm 1.8$ | $7.0 \pm 3.2$ |
| **Braingle Logic** | | | | |
| OpenAI o3 Solution | $8.1 \pm 3.8$ | 7.0 | $2.2 \pm 1.8$ | $5.8 \pm 3.2$ |
| DeepSeek V1 Solution | $12.3 \pm 5.2$ | 11.0 | $3.3 \pm 2.5$ | $9.0 \pm 4.6$ |
| Llama 70B Solution | $11.0 \pm 4.6$ | 10.0 | $2.9 \pm 2.1$ | $8.1 \pm 4.3$ |

Table S3: Mean and standard deviation of solution step counts for the 30 most difficult puzzles in the Braingle Math and Braingle Logic datasets. (*For math, 1 human solution and 1 model solution were excluded as outliers due to excessive casework, 1 human solution excluded due to directly asserting answer)

| | Solution Steps | Median | Creative Steps | Rudimentary Steps |
|---|---|---|---|---|
| **Braingle Math – OpenAI o3 Step Count** | | | | |
| Human Solution $(n = 28)^*$ | $7.8 \pm 3.5$ | 7.0 | $2.1 \pm 1.4$ | $5.7 \pm 2.6$ |
| OpenAI o3 Solution $(n = 29)^*$ | $7.7 \pm 2.9$ | 8.0 | $2.0 \pm 1.0$ | $5.7 \pm 3.0$ |
| OpenAI o3 Correct Solution $(n = 21)$ | $8.0 \pm 2.2$ | 8.0 | $1.9 \pm 0.9$ | $6.1 \pm 2.4$ |
| **Braingle Math – Deepseek R1 Step Count** | | | | |
| Human Solution $(n = 28)^*$ | $7.8 \pm 4.7$ | 7.0 | $3.9 \pm 2.3$ | $3.9 \pm 2.7$ |
| OpenAI o3 Solution $(n = 29)^*$ | $6.8 \pm 3.2$ | 6.5 | $3.3 \pm 1.3$ | $3.5 \pm 2.5$ |
| OpenAI o3 Correct Solution $(n = 21)$ | $7.1 \pm 3.1$ | 7.0 | $3.2 \pm 1.3$ | $4.0 \pm 2.5$ |
| **Braingle Logic – OpenAI o3 Step Count** | | | | |
| Human Solution $(n = 30)$ | $10.6 \pm 5.2$ | 9.0 | $2.9 \pm 1.7$ | $7.8 \pm 5.5$ |
| OpenAI o3 Solution $(n = 30)$ | N/A | N/A | N/A | N/A |
| OpenAI o3 Correct Solution $(n =??)$ | N/A | N/A | N/A | N/A |
| **Braingle Logic – Deepseek R1 Step Count** | | | | |
| Human Solution $(n = 30)$ | $9.6 \pm 4.7$ | 9.0 | $5.3 \pm 2.3$ | $4.3 \pm 2.6$ |
| OpenAI o3 Solution $(n = 30)$ | - | - | - | - |
| OpenAI o3 Correct Solution $(n =??)$ | - | - | - | - |

systematic exploration of the option space. This may be an issue with prompting, or could be the fact that DeepSeek may be associating significant progress with creativity rather than the innovation of the ideas used. Then, step 8 from DeepSeek is declaring the final answer that should have been obtained from step 7, so this should be the most rudimentary because no additional deduction was made. It is interesting to see that DeepSeek R1 considers this as creative because it synthesizes the results, but after observing other DeepSeek responses, we see that calling the declaration of the final result a creative step is not a common error and that erroneously associating significant progress rather than innovation with creativity may be a more likely source of why DeepSeek R1 categorizes a greater number of steps as creative than OpenAI o3.

From Table S2, we observe that DeepSeek V1 solutions have more steps than Llama 70B solutions for both math and logic. OpenAI o3 outputs have fewer steps than both of the other models for both math and logic brainteasers. We also observe that logic problems generally have more steps than math problems as seen with DeepSeek V1 and Llama 70B solutions; that said, for OpenAI o3, the average step count for logic problems is marginally smaller. The DeepSeek V1 and Llama 70B responses are both considerably longer than human solutions, which may be a result of continued trial-and-error attempts that would not be noted in human solutions.

We now present an example of an incorrect solution from DeepSeek V1 followed by the OpenAI o3 step breakdown. As the DeepSeek solution is very long, some portions are truncated but the overall sequence of logic is still present.

First, let's clarify some terms to ensure we're on the same page.

## Understanding the Terms

1. **Palindrome**: A palindrome is a number (or word) that reads the same backward as forward. For example, 121, 1331, and 1 are palindromic numbers.

2. **Non-palindromic integer**: An integer that is not a palindrome. For example, 12, 123, and 10 are non-palindromic, since reading them backward gives 21, 321, and 01, respectively.

3. **Cube of an integer**: The integer multiplied by itself three times. For example, the cube of 3 is
$$3^3 \ = \ 3 \times 3 \times 3 \ = \ 27.$$

## The Problem Restated

We're looking for an integer $n$ such that $n$ is *not* a palindrome, $n^3$ is a palindrome, and $n$ is the only such integer.

## Checking Small Integers

1. $n = 1$ :
   - 1 is a palindrome, $1^3 = 1$, which is a palindrome, excluded since $n$ itself is palindromic.

2. $n = 2$ :
   - 2 is a palindrome, $2^3 \ = \ 8$, which is a palindrome, excluded since $n$ is palindromic.

   $\vdots$

3. $n = 26$ :
   - 26 is not a palindrome, $26^3 = 17576$, which (by the problem's assertion) is a palindrome, this is the unique solution.

## Considering Negative Integers

A negative integer is non-palindromic if its positive counterpart is not a palindrome (since the negative sign makes it non-palindromic, as palindromes are typically considered for absolute values in numbers).

Let's check some negative integers. (The model then checks -1, -2, -10, and -11, but gives up after finding no success).

Perhaps the answer is among positive integers, and I missed it earlier.

## Re-evaluating Positive Integers

Let me think about the cube being a palindrome. The cube must read the same backward as forward.

Looking back, the only cubes that are palindromes from the numbers I checked are 1 (from 1), 8 (from 2), 343 (from 7), and 1331 (from 11).

But all these 'n' values are palindromic themselves.

Wait, is there any non-palindromic 'n' whose cube is a palindrome? Maybe higher numbers? (The model then tries 101 and 1001 but fails as 101 and 1001 are palindromes.

## Model Response: Math 11 (DeepSeek V1) continued

This is perplexing. Maybe the answer is that there is no such non-palindromic integer whose cube is a palindrome. But the problem states that there is one, implying it exists.

Wait, perhaps I missed a small number. Let me check 'n = 0'. 0 is a palindrome, $0^3 = 0$, which is a palindrome, but 0 is palindromic.

Alternatively, maybe the answer is 10, but earlier we saw $10^3 = 1000$ isn't a palindrome. Unless we consider 1000 as a palindrome if we ignore trailing zeros, but that's not standard.

After some research or recalling, I think the answer is **26**, but earlier calculation showed $26^3 = 17576$, which isn't a palindrome. Maybe I miscalculated.

Let me calculate $26^3$ carefully. (The model obtains that $26^3 = 17576$. 17576 backward is $67571 \neq 17576$. So not a palindrome.

(The model then continues to try numbers it has already tried out, such as 0, 10, -1, and 26, but still is unsuccessful.)

⋮

Given that, and since the problem states there is one, the most likely answer is **26**, accepting that 17576 is a palindrome (perhaps by some definition where the middle digit is the center and the others mirror, but they don't here).

However, standard definition says a palindrome reads the same backward, which 17576 does not ($67571 \neq 17576$).

Given this confusion, and based on mathematical sources, the accepted answer is:

**26**, with $26^3 = 17576$ being considered a palindrome, even though by strict digit reversal it's not. This might be a special case or an exception in how palindromes are defined for this problem.

## Model Response: Math 11 (OpenAI o3 Breakdown)

1. State and explain the relevant definitions of "palindrome," "non-palindromic integer," and "cube," establishing the criteria the sought integer must satisfy. (Rudimentary)
2. Propose a brute-force method: examine successive integers, compute each cube, and check both n and n³ for palindromicity. (Rudimentary)
3. Implement the method for n = 1–9; record that either n is itself a palindrome or n³ is not, so none of these values fits the requirement. (Rudimentary)
4. Continue the same manual search for n = 10–22 and again find no qualifying integer. (Rudimentary)
5. Rely on a recollection/intuition and single out n = 26 as a promising candidate instead of continuing the systematic scan in order. (Creative)
6. Calculate 26³ = 17 576 and test 17 576 for palindromicity, concluding (incorrectly) that it is a palindrome. (Rudimentary)
7. Declare that, because 26 appears to work and no other candidates have been discovered, 26 must be the unique non-palindromic integer whose cube is a palindrome, and give 26 as the final answer. (Rudimentary)

From reading the outputs, it is clear that OpenAI o3 maintains the same chain of thought as the DeepSeek V1 response while consolidating much of the trial and error performed on previously tested candidates. Like DeepSeek V1, the OpenAI o3 also notes that 17576 is not a palindrome. We also find that the steps are larger components of the problem rather than just individual deductions as desired, though we could argue that steps 3 and 4 could be combined since the method of search is identical in both steps. Nonetheless, we still see that OpenAI o3 breaks down a provided model

Table S4: An example of creative and rudimentary steps.

| Creative Step Example | Rudimentary Step Example |
| --- | --- |
| Examine the given input-output pairs and discover the hidden rule: write each number in binary and weight each digit—count 1 for every 0 and 2 for every 1. | Convert 9304 to binary, obtaining 10010001011000. |

response into macro-steps that correctly highlight the chain of thought even when the response can yield an incorrect answer. This shows that the OpenAI o3 step breakdowns do not look beyond the provided response itself and maintain the thought process given by the response.

One drawback of using the step count to measure the complexity of a problem is that the complexity may vary significantly between steps. For many problems, there are fewer insights, but the insights are more difficult to obtain and are more significant in solving the problem; yet the model's presentation of these steps does not necessarily highlight the differences in complexity between steps. This is particularly true for math problems that ask for some common pattern between a list of numbers without giving any hint of the pattern. Without acknowledging the required trial-and-error and time needed to discover the step's key insight, such "steps" would be considered equivalent in complexity to routine arithmetic calculations under this measurement. Therefore, we also make the distinction between creative steps and rudimentary steps. In the problem that provides a set of input-output pairs where the goal is to find the corresponding output of 9304, we show in Table S4 that finding the input-output rule is considered creative as it significantly reduces the difficulty of the problem, while carrying out the rule on converting the designated number 9304 is rudimentary since calculations like binary conversion are simple and routine. The explicit steps themselves are provided in Supplementary Section G.

# H   Braingle Math Dataset Categorization

## H.1   Overview

### H.1.1   Categories

We manually categorize the math dataset based on problem style, into the following three categories and respective subcategories. Category (1) represents problems which could reasonably be found on a standard math competition such as AMC, AIME, etc. Category (2) represents problems which are still mathematically rigorous, but do not quite fit the style of a competitive math problem. Category (3) represents problems which are not rigorous, and involve finding patterns or thinking outside the box.

1. **Standard competitive math:**

   - **Geometry -** Problem with spatial reasoning, finding areas/side lengths, etc.
   - **Number Theory -** Divisibility, factorization, etc.
   - **Combinatorics -** Counting, probability, etc.
   - **Algebra -** Setting up and solving systems of real-valued equations

2. **Nonstandard:**

   - **Logic -** Problems resembling mathematical Logic puzzles
   - **Special Number -** Finding a "mystery" number with desired properties

3. **Heuristic/non-rigorous:**

   - **Pattern -** Finding the "next number" or a common rule for a group or sequence of numbers
   - **Arithmetic -** Rearranging an arithmetic expression to achieve a target number

## H.2 Examples

### H.2.1 Standard category

> **Problem: Math 50 (Geometry)**
>
> It's easy to see that a ring can completely hold (surround) two identical smaller rings with half the diameter, without overlapping. Three times the diameter, the bigger ring is space enough to seat seven rings; the outer six touching both the middle ring and the bigger circle/perimeter. Using this basic information and your imagination, determine the maximum number of rings that could be housed inside another ring with four, five, six and seven times the diameter.

Geometry problems require spatial reasoning– in this case, the solver must visualize an arrangement of circles contained within a larger circle, and argue about the "total diameter" of this arrangement.

> **Problem: Math 124 (Number Theory)**
>
> I have a machine which has four cog wheels in constant mesh. The largest cog has 242 teeth and the others have 160, 64 and 22 respectively. How many revolutions must the largest cog make before each of the cogs is back in its starting position?

Number theory problems often involve concepts like common divisors, multiples, and remainders.

> **Problem: Math 117 (Combinatorics)**
>
> What is the largest number of pieces you can form with $n$ straight cuts of a pizza? The pieces do not need to be of equal size.

While this may seem like a problem requiring a visual arrangement, a clever counting argument can simplify the problem greatly: for every cut added, if it intersects $k$ of the existing cuts, it adds $k + 1$ new pieces to the arrangement.

> **Problem: Math 125 (Algebra)**
>
> Three people (A, B, and C) need to cross a bridge. A can cross the bridge in 10 minutes, B can cross in 5 minutes, and C can cross in 2 minutes. There is also a bicycle available and any person can cross the bridge in 1 minute with the bicycle. What is the shortest time that all men can get across the bridge? Each man travels at their own constant rate.

Algebra problems often involve word problems with times, prices, and rates; they can be easily computed after a proper mathematical setup from the word problem.

### H.2.2 Nonstandard category

> **Problem: Math 122 (Logic)**
>
> The people of Olde Mathville had unique ways of punishing wayward citizens. For example, those convicted of crimes of dishonesty were chained to the Liars' Rail until they solved a number of puzzles.
>
> One such puzzle has been recently discovered! In the multiplication below, each letter - L, I, A, R, and S - takes the place of a different digit. Find the digits to make the multiplication true.
>
> L I A R
> x S
> ─────
> R A I L

While this problem is still mathematically themed, the primary focus of the problem is in logical reasoning– the solver must assign five unique values to the five unknown entities, as one would in a logic grid.

---

**Problem: Math 75 (Special Number)**

What is the smallest number that is the sum of two different pairs of cubes?

---

In this category, the solver must find a number or set of numbers satisfying a unique set of constraints. This category can be seen as adjacent to **Number Theory**; however, in **Special Number** puzzles, the primary challenge is not in making rigorous number-theoretic arguments relating to divisibility, but rather deductive, heuristic, and sometimes "lucky" observations.

### H.2.3 Heuristic category

---

**Problem: Math 79 (Pattern)**

What is so special about this particular sequence of numbers?
425260376469080434957

---

Here, the problem does not have a rigorous final answer, but rather requires general pattern-finding. In this problem, the listed numbers are the digits of pi, with 1 added to each. We review all such problems to ensure that the final answer is both "reasonable" and the unique "reasonable" solution (i.e. no other plausible patterns exist).

---

**Problem: Math 216 (Arithmetic)**

Add the appropriate mathematical operators or symbols to make the following correct:
3 4 5 = 90
Parentheses may be used freely.

---

In these problems, the solver inserts mathematical operators to achieve a desired numerical goal. We place these kinds of problems in the **Heuristic** category, as the solver is often required to think outside the box, and the set of valid "actions" is not always rigorously defined. In this case, the answer requires use of a factorial:

$$3/4 \times 5! = 90$$

## I Braingle Logic Dataset Categorization

### I.1 Overview

### I.1.1 Metrics

For each brainteaser, where applicable, we evaluate the following metrics:

- **Depth:** The number of distinct traits in each "grouping"
- **Width:** The number of "groups"
- **State Space Size:** The total number of possible arrangements within the initial defined constraints of the problem
- **Number of clues:** The number of distinct pieces of information given, AFTER the state space is defined

### I.1.2 Categories

We manually categorize the logic dataset based on general structure of each problem, into the following four categories and respective subcategories. Note that categories (1) and (2) represent rigorous logical deduction problems with bounded state spaces. In (3), problems do NOT have well-defined or bounded state spaces, with answers ranging from numbers to descriptions of algorithms. In (4), answers are not logically rigorous, but can still reflect a model's pattern-finding or heuristic reasoning skills.

- **(1) Large state space, simple clues:**
  - **0D Logic Grid -** Standard logic grid with no positional reasoning. $d$ traits, $w$ options per trait
  - **1D Positional -** Puzzle with 1D positional reasoning. Includes races, seating arrangements, etc.
  - **2D Positional -** Puzzle with 2D positional reasoning. Includes Bingo cards, chess boards, etc.
  - **Number -** Guess the number; deduction-style problem with mathematical clues. $d = 2$, $w$ digits
  - **Clusters -** No fixed traits, cluster $wd$ objects freely into $w$ groups of $d$
  - **Tree -** Puzzle with tree-like structure. Includes family trees, elimination-style tournaments.
- **(2) Small state space, complex clues:**
  - **Liars -** Clues themselves may be true or false
  - **Communication -** Impartial information between multiple parties, typically through conversation
  - **Compound -** Multi-claused or conditional clues (i.e. many if, then, or statements)
- **(3) Math-like:**
  - **Algorithm -** Design and/or execute an algorithm
  - **Math -** Similar to comp-math, unbounded numerical state space
- **(4) Heuristic/non-rigorous:**
  - **Pattern -** Observing a non-rigorous pattern, "riddle"-like problems
  - **Linguistic -** Clues based on semantic meaning

## I.2 Examples

### I.2.1 Computing metrics

A statement defining "state space" is a statement which gives general information about the problem, which is standard among other problems of the same type. This may include defining traits (name, occupation, age), defining options per trait ("occupations are welder, bricklayer, ..."), setting number of items per category ("there are 2 welders, 1 bricklayer..."), and declaring other constraints (no repeating digits). In contrast, a "clue" is defined as a piece of information which is unique to the given problem. Note that for some problems, the concept of "depth" and "width" can be up to interpretation. These metrics are made to generally quantify the difference in the various categories.

Here is an example solution for a classic logic grid-style puzzle:

---

**Problem: Logic 15**

1st House: Yellow, Norwegian, Water, Cats, Dunhill
2nd House: Blue, Dane, Tea, Horse, Blends
3rd House: Red, Brit, Milk, Birds, Pall Malls
4th House: Green, German, Coffee, FISH, Prince
5th House: White, Swede, Beer, Dogs, Bluemasters

---

Each "grouping" contains 6 traits: house, color, nationality, beverage, pet, and cigar, so the **depth** is 6. There are 5 groupings, so the **width** is 5. For a classic logic grid with depth $d$ and width $w$, the **state space** is given by: $(w!)^{d-1} = (5!)^{6-1}$. Note that we define state space as the entire required arrangement, NOT just the space of possible final answers.

---

**Problem: Logic 3**

There is a ten-digit mystery number (no leading 0), represented by ABCDEFGHIJ, where each numeral, 0 through 9, is used once. Given the following clues, what is the number?
1) A + B + C + D + E is a multiple of 6.
2) F + G + H + I + J is a multiple of 5.
3) A + C + E + G + I is a multiple of 9.
4) B + D + F + H + J is a multiple of 2.
5) AB is a multiple of 3.
6) CD is a multiple of 4.
7) EF is a multiple of 7.
8) GH is a multiple of 8.
9) IJ is a multiple of 10.
10) FE, HC, and JA are all prime numbers.

---

Here, a "grouping" is simply pairing digit positions with digits, giving a **depth** of 2. There are ten positions, giving a **width** of 10. There are ten **clues** (not counting the initial set up information), and the **state space size** is given by $9 \cdot 9!$, accounting for the non-leading zero information.

### I.2.2 Large/simple category

---

**Problem: Logic 32 (0D)**

These five businessmen represented different companies at a recent trade fair. Unfortunately, the hotel they were all due to stay in had accidentally double booked their rooms. They tried other hotels in the area, but all were fully booked. Consequently, they all agreed to share the only two rooms available in the hotel - one twin and one triple. From the clues, can you work out each man's name, company and official title?
1. When the five men realized their dilemma, they drew straws to see who would share with whom. The outcome for four of them was that the CEO shared with the businessman from ABM Inc. and Edgar shared with the Developer.
2. Alan does not work for Reed Right and is not the Director. The Director does not work for Lantel or Blue Teeth.
3. Neither Clarkson nor Grimaldi works for Reed Right. Grimaldi is either the IT Analyst or the CEO.
4. The businessman from Reed Right did not share a room with Thomas.
5. Clarkson, who does not work for Lantel, shared with either Thomas or Grimaldi but not both.
6. Edgar's surname is either Casson or Graves. Edgar did not share with Carl.
7. The businessman from Chiptech shared with the businessman from Lantel.
8. The CEO shared with the IT Analyst.
9. Casson works for either Blue Teeth or Chiptech.
10. Neither Joshua nor Alan works for ABM Inc.
11. Fielder shared with the Director.
12. The Accountant from Blue Teeth shared with the Developer.
First names: Alan, Carl, Edgar, Joshua, Thomas.
Last names: Casson, Clarkson, Fielder, Graves, Grimaldi.
Company: ABM Inc., Blue Teeth, Chiptech, Lantel, Reed

---

While there is some variation, the majority of puzzles in the 0D category follow the standard logic grid format, where we are given a fixed set of categories (first names, last names, companies), and a set of clues which each give a fairly simple individual deduction.

---

**Problem: Logic 178 (1D)**

List the order in which each person finished.
Tommy Tombstone finished after Lance Lamers and Brett Brown but before Mitch Monday.
Peter Poultry finished before Daniel Dusk and Lance Lamers.
Sam Sunny finished after Peter Poultry and before Jack Jill and Harry Hills.
Keri Kernel finished after Peter Poultry, Mitch Monday and Tommy Tombstone.
Lance Lamers finished after Brett Brown and Daniel Dusk, but before Jack Jill and Mitch Monday.
Mitch Monday finished after Sam Sunny and Brett Brown.
Brett Brown finished before Jack Jill, Mitch Monday and Peter Poultry.
Daniel Dusk finished before Keri Kernel and Tommy Tombstone, but after Sam Sunny.
Jack Jill finished before Keri Kernel, Tommy Tombstone and Mitch Monday, but after Peter Poultry and Daniel Dusk.
Harry Hills finished before Mitch Monday but after Lance Lamers, Jack Jill and Tommy Tombstone.

---

A race is a standard way to express a 1D problem. While these kinds of problems can be reduced to the forms of logic grids (by considering "position" as a category), problems of this type generally possess higher width and lower depth, as there is higher emphasis on ordering a large number of characters.

## Problem: Logic 12 (2D)

You are given a stack of bingo cards. Your task is to find a specific card. Given the following clues, what is the number arrangement of that card?

Columns, left to right, are: B (contains numbers 1 through 15), I (contains numbers 16 through 30), N (contains numbers 31 through 45), G (contains numbers 46 through 60), O (contains numbers 61 through 75). Rows, top to bottom, are: 1, 2, 3, 4, 5. An example of coordinate nomenclature: B1 identifies column B row 1. N3 is a free space (contains no number). No number appears more than once.

1) Each numeral (0 through 9) appears one time in Row 1.
2) The sum of the numbers in Row 4 is a square number.
3) There is only one two-digit prime number in each row.
4) The range of the numbers in Column N is 8.
5) Each number in Column G has a tens digit that is less than the units digit.
6) Each number in Column O is odd.
7) In only one column are the numbers in descending order from top to bottom.
8) Each column has only one numeral that appears exactly two times.
9) The smallest number in each column is located in Row 5.
10) The sums of each column share a single common prime factor.
11) The numeral 5 only appears one time on the card.
12) The sum of the numbers in each diagonal is an odd number.
13) The product of B3 and O3 has a units digit of 2.
14) The product of I3 and G3 has a units digit of 4.

There are four bingo-style questions in the Logic dataset. Here, the solver must deduce the values of an entire 5x5 grid, where clues may pertain to columns, rows, diagonals, or individual cells. These problems tend to have incredibly large state spaces, but clues can be very reductive (e.g. clue 6 in this puzzle reduces the state of possible answers by a factor of around $2^5$ on its own).

## Problem: Logic 10 (Number)

Professor Abacus is purchasing a ticket for the Deca Lotto. The lotto number has ten digits, using the numerals 0 through 9, each numeral used once. The clerk asked what number he wanted to pick. Professor Abacus handed the clerk a piece of paper with nine statements, saying ""If you can correctly figure out the number, I will give you half of whatever I win.""
What is the number?

1) The sum of the first five digits is a prime number.
2) The sum of the last five digits is a triangle number*.
3) The sum of the digits in the odd positions is an odd number.
4) The sum of the middle two digits is a square number.
5) The sum of the middle four digits is a cube number.
6) The difference between the 1st and 10th digits is two.
7) The difference between the 2nd and 9th digits is three.
8) The difference between the 3rd and 8th digits is four.
9) The numeral 4 is somewhere in the first five positions.
* You can form a triangle arrangement by building it in the pattern row 1 = 1, row 2 = 2, row 3 = 3 etc. eg. 10 is a classic triangle number as per ten pin bowling. They are arranged in a triangle 1, 2, 3, 4.

The second example shown in I.2.1 is also a classic number problem. As with the 1D problems, Number problems can be reduced to logic grids, where we are grouping "digits" with "positions". However, Number problems are distinct in the mathematical nature of their clues, which cannot be replicated nicely in natural language form, where the digits and positions are replaced with arbitrary names and placeholders.

> **Problem: Logic 35 (Clusters)**
>
> A child has 4 blocks with a different letter on each side (no letter is repeated on different blocks either). If the list of words below can all be formed using these blocks, figure out which letters belong on which blocks...
>
> skid, hoax, joey, glum, rand, grit, monk, fair, vane, wide, cafe, dupe, joke, bail, shop

In this problem, we must form four unordered clusters of 6 letters each. These problems differ from traditional logic grids in the sense that individual category names are no longer specified– rather than grouping across six different categories, all entities belong to one category: letters.

> **Problem: Logic 81 (Tree)**
>
> One day, a college student named Tina walked into her logic class and waited for her fun day of logic once again. Finally the professor, Professor C. D. Rock, walked in saying, "I just went through some stuff and stumbled upon an old family tree, that belonged to my grandparents, that gave me an idea. I have here a few clues, and you have to use them to figure out their family tree!"
>
> Tina then gets the clues, and tries to work them out. Unfortunately this puzzle is a little harder than the normal ones Professor C. D. Rock gives out. Can you help her?
>
> The family tree consists of two grandparents, who had 3 children, each of whom get married and have 2 children.
>
> Males: Cole, Cristian, Jason, Neil, and Steve
>
> Females: Amanda, Ashley, Beth, Erin, Kaitlyn, Katherine, Makayla, Payton, and Tammy
>
> Clues:
>
> 1. One of Makayla's cousins is Jason's son.
> 2. One of Ashley's aunts is Tammy.
> 3. Tammy's brother-in-law is Neil's son.
> 4. Kaitlyn's sister is Ashley's cousin.
> 5. Ashley's uncle, Steve, is Erin's brother-in-law.
> 6. The three uncles are Payton's dad, Cristian, and Katherine's son.
> 7. The three aunts are Kaitlyn's mom, Ashley's mom, and Cristian's sister-in-law.
> 8. Jason's brother is Ashley's dad.
> 9. Amanda's sister is Steve's niece.
> 10. Beth is not Cole's aunt.

In Tree problems, we need to fill in the identities of a family tree– this requires hierarchical thinking, where deductions about a person's "generation" intersect deductions about a person's individual relationships.

### I.2.3 Small/complex category

> **Problem: Logic 85 (Liars)**
>
> Edward, Howard, and John are three high school students each of whom is taking three of the four subjects, biology, chemistry, history, and mathematics. One day while talking about their programs they made the following statements.
>
> Edward: There is just one subject we're all taking. I'm the only one of us who is taking mathematics. No two of us are taking the same three subjects. John is wrong when he says that Howard and I are both taking chemistry.
>
> Howard: Ed is the only one of us who is taking history. John and I are taking the same subjects. We're all taking biology. Two of us are taking both chemistry and biology.
>
> John: We're all taking mathematics. Howard is taking history. Ed is taking one subject that I'm not. Both Howard and Ed are taking chemistry.
>
> If two and only two of each boy's statements are true, what subjects is each boy taking?

The state space of this problem is very small– for each student, one must deduce which subject they are *not* taking, out of 4. In Liars problems, complexity arises from the fact that clues themselves may be true or false, and unlike problems in the **Large/simple** category, immediate deductions cannot be

easily made from each clue. Such problems are susceptible to brute-force testing; a model can easily test every possible arrangement of subjects, and check how many statements are true or false, until exactly two of each boys' statements are true.

---

**Problem: Logic 104 (Communication)**

Mr. Simkin, the new math teacher at school, was impressed by his students' ability to solve logic puzzles. He pulled aside three more students, and handed them each a sealed envelope with a number written inside. He told them that they each have a positive integer, and the sum of their numbers was 14.

Manny, Moe, and Jack each opened their envelopes. Mr. Simkin asks Manny if he knows anything about the numbers the other two are holding, and Manny says, "I know that Moe and Jack are holding different numbers."

Moe answers, "IN THAT CASE, I know that all three of our numbers are different."

Jack thinks for a bit, and then says, "Now I know all of our numbers."

Mr. Simkin turns to the class and asks if anyone in the class knows the numbers. Gretchen's hand shoots up into the air, and after waiting for a while to see if anyone else will get the answer, Mr. Simkin calls on Gretchen.

What numbers does she say they each are holding?

---

In Communication problems, the reader must make deductions based on the deductions of multiple agents with impartial information. This dynamic adds inherent complexity to each clue; rather than only considering the deductions the reader themselves can make, the reader must also keep track of the deductions that each character can individually make as well.

---

**Problem: Logic 235 (Compound)**

Just before the end of the term four high school students were discussing their chances for certain grades. The following remarks contain the gist of their hopes and fears.

Jack: We'll all get different grades. If I get an A, then Lucy will get a D.

Jean: If Lucy gets a C, then Jack will get a D. Jack will get a better grade than Paul.

Lucy: If Jean doesn't get an A, then Jack will get a C. If I get a B, then Paul won't get a D.

Paul: If Lucy gets an A, then I'll get a B. if Jean doesn't get a B, I won't either.

When the final examinations were graded and the term marks made out each of the four passed, and strange as it may seem, each received a grade that checked exactly with all the ideas they had previously expressed.

What grade did each receive?

---

Here, each individual clue is a conditional statement, and only allows for very specific deductions (e.g. Jack has an A $\implies$ Lucy has a D). Once again, we see that problems of this type are susceptible to brute force: there are only 4! arrangements of grades for the four students, and a model can easily iterate through all arrangements and check for contradictions in the logical statements.

### I.2.4 Math-like category

---

**Problem: Logic 38 (Algorithm)**

It is your task to determine how high you can drop a billiard ball without it breaking. There is a 100 story building and you must determine which is the highest floor you can drop a ball from without it breaking. You have only two billiard balls to use as test objects. If both of them break before you determine the answer then you have failed at your task. What is the order of floors that you should drop the balls from to minimize the number of droppings that you will have to make to determine the answer?

Assume that if a ball doesn't break you can reuse it without worrying about it being weakened.

---

In this problem, the reader must *design* an algorithm to systematically determine highest "droppable" floor. In some Algorithms problems, the reader may simply have to *execute* a series of actions.

> **Problem: Logic 171 (Math)**
>
> Mad Ade's Great Uncle Gaseous O' Windpants owned the Madadian Grocery store "The Beggars Can't Be Choosers". Madadia was well renowned for its pungent cheeses, especially its "Kebabrie" and "Chillirella".
> On display in the store were six cheese pieces weighing 15, 16, 18, 19 , 20 and 31 Pounds.
> Five out of the six pieces are "Kebabrie" and the remaining one is "Chillirella".
> Norma Leigh-Sobar purchased two pieces of "Kebabrie" and Laura Anne-Hardy also purchased some "Kebabrie", but she purchased twice as much in weight than Norma.
> How much does the remaining "Chillirella" weigh?

Here, we have a classic algebra problem framed as a logic puzzle, where the reader must set up a system of equations to solve.

### I.2.5 Heuristic category

> **Problem: Logic 82 (Pattern)**
>
> I am a word of five letters. Multiply my fifth by two and you have my first. Divide my first by twenty and you have my third. Divide my third by five and you have my second or fourth.

Pattern problems often take the form of "riddles": there is no rigorous or clear problem statement, and no list of deductions. In this case, the reader must think outside of the box, thinking of letters as Roman numerals (the answer to this one is "CIVIL").

> **Problem: Logic 44 (Linguistic)**
>
> In the following code, each symbol stands for one of five letters.
> ! stands for T, E, O, Z, or Y
> ? stands for F, G, A, Q, or I
> # stands for N, I, W, A, or U
> $ stands for T, E, N, I, or H
> < stands for R, C, A, S, or B
> * stands for I, D, E, U, or S
> & stands for J, I, E, P, or K
> ˆ stands for O, L, G, I, or H
> > stands for L, S, N, C, or E
> The nine letter code word, <$ˆ&?*!#>, can be translated into two English words that are opposites. What are the two words?

Like standard logic puzzles, the state space is well-defined; the reader has a list of options for each letter position. However, the deductions in this case are entirely heuristic– the only "clue" the reader has is the fact that the two 9-letter words are semantic opposites. Much of deduction is based on general word sense (common consonant combinations, vowel arrangements, etc.).

### I.3 Full Population Statistics

In Table S5, we provide a complete version of the abbreviated table shown in 1b. Here, "Log state space" is calculated in base 10.

Table S5: Full population statistics for the Logic set, by category.

| Category | Count | Depth | Width | Log state space | Clues | Popularity | Difficulty |
|---|---|---|---|---|---|---|---|
| **Simple/large** | | | | | | | |
| 0D | 29 | $3.79 \pm 1.26$ | $5.41 \pm 1.38$ | $6.9 \pm 3.29$ | $8.29 \pm 4.02$ | $2.53 \pm 0.23$ | $2.68 \pm 0.23$ |
| 1D | 13 | $2.92 \pm 1.26$ | $7.85 \pm 2.64$ | $7.04 \pm 3.83$ | $9.15 \pm 3.02$ | $2.56 \pm 0.3$ | $2.6 \pm 0.28$ |
| 2D | 22 | $2.62 \pm 1.43$ | $26.81 \pm 23.43$ | $14.95 \pm 9.38$ | $9.38 \pm 6.7$ | $2.47 \pm 0.3$ | $2.72 \pm 0.25$ |
| Number | 17 | $2.12 \pm 0.49$ | $8.94 \pm 1.82$ | $6.47 \pm 0.93$ | $5.94 \pm 2.59$ | $2.43 \pm 0.27$ | $2.74 \pm 0.26$ |
| Clusters | 8 | $5.5 \pm 1.51$ | $5.25 \pm 3.15$ | $12.38 \pm 3.43$ | $12.38 \pm 3.29$ | $2.56 \pm 0.2$ | $2.78 \pm 0.13$ |
| Tree | 6 | $2.4 \pm 0.55$ | $10.6 \pm 3.29$ | $6.6 \pm 1.92$ | $9.2 \pm 2.95$ | $2.83 \pm 0.17$ | $2.69 \pm 0.15$ |
| **Complex/small** | | | | | | | |
| Liars | 17 | $2.25 \pm 0.58$ | $6.06 \pm 2.72$ | $2.26 \pm 1.76$ | $8.65 \pm 3.22$ | $2.5 \pm 0.28$ | $2.66 \pm 0.22$ |
| Communication | 4 | $2 \pm 0$ | $3.67 \pm 1.15$ | $1.93 \pm 1.16$ | $3.67 \pm 0.58$ | $2.7 \pm 0.28$ | $2.55 \pm 0.15$ |
| Compound | 9 | $2.33 \pm 0.71$ | $5.78 \pm 3.42$ | $3.35 \pm 2.45$ | $7.78 \pm 2.95$ | $2.48 \pm 0.44$ | $2.67 \pm 0.34$ |
| **Math-like** | | | | | | | |
| Algorithm | 38 | - | - | - | - | $2.56 \pm 0.34$ | $2.66 \pm 0.23$ |
| Math | 32 | - | - | - | - | $2.41 \pm 0.27$ | $2.63 \pm 0.22$ |
| **Heuristic** | | | | | | | |
| Pattern | 26 | - | - | - | - | $2.48 \pm 0.27$ | $2.63 \pm 0.22$ |
| Linguistic | 15 | $2 \pm 0$ | $7.64 \pm 3.77$ | $7.15 \pm 2.35$ | $1.93 \pm 2.58$ | $2.5 \pm 0.12$ | $2.61 \pm 0.15$ |

# J    Model Performance by Categories

Here we display complete visuals on model performance by category and subcategory, in both Math and Logic datasets, as defined in Section 3.2. We display results for model performance, subset by category, in Figures S1 and S2. We display the same results, further divided into subcategories, in Figures S3 and S4.

In both the Math and Logic datasets, models consistently perform the poorest in the shared **Heuristic** category. Despite the Heuristic categories attaining a similar human difficulty rating to the average in their respective datasets, language models seem to disproportionately struggle in this pattern-focused category, suggesting a relative lack of creative reasoning skills.

In the Logic dataset, models tend to perform more poorly in the **Simple/large** category when compared to the **Complex/small** category; thus, compared with humans, language models struggle more with large volume, and less with complex logical statements. This may be caused by two factors: (1) models inherently struggle with long, sequential chains of reasoning, and (2) models rely on brute-force methods to solve puzzles, which are feasible for Complex/small puzzles but do not work on Simple/large puzzles with larger state spaces (see Section I.2.3).

Finally, across both datasets, it remains evident that models struggle with spatial reasoning. In the Math dataset, this is evidenced by low model performance in the **Geometry** subcategory, and in the Logic dataset, this is evidenced by low model performance in the **2D** subcategory.

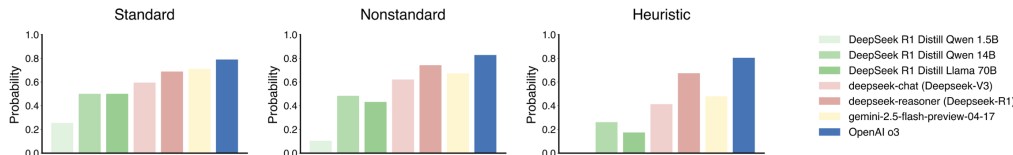

Figure S1: Correctness on Math categories, using the `Math` prompt.

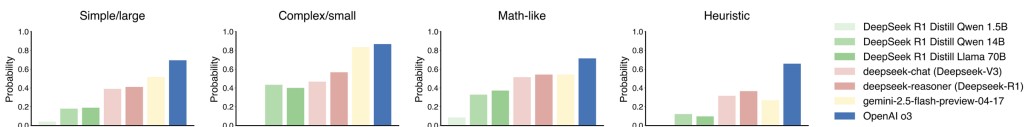

Figure S2: Correctness on Logic categories, using the `Math` prompt.

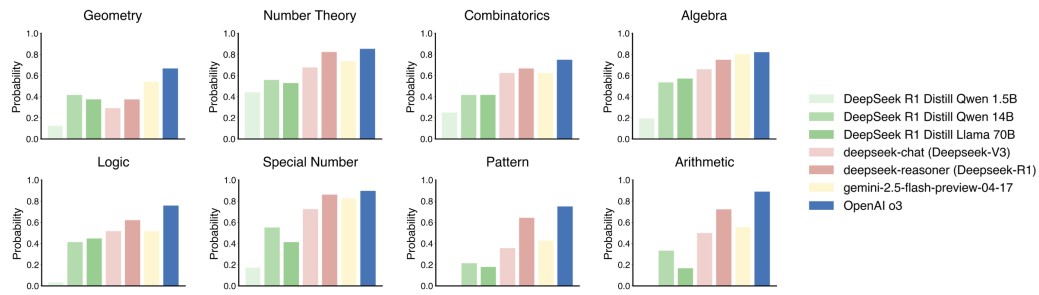

Figure S3: Correctness on Math subcategories, using the `Math` prompt.

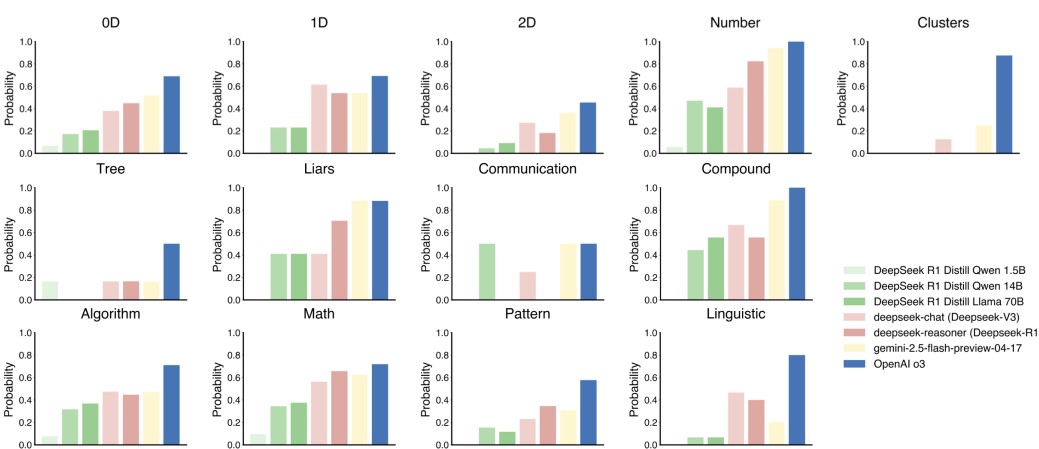

Figure S4: Correctness on Logic subcategories, using the `Math` prompt.

# K Brute-Force

## K.1 Brute Force Prompting Analysis

Here is an example problem and response excerpt when OpenAI o3 is given no additional prompting:

---

### Problem: Math 20

Sandy and Sue each has a whole number of dollars. Sandy says: "If Sue gives me some dollars, we'll have the same amount. But if I give Sue the same number of dollars, she'll have twice as much money as I have." Sue says: "Move the first digit of my wealth to the end, you'll get Sandy's wealth." If they both have at most 1 million, how much do they each have?

---

### Model Response: Math 20 (No Prompting)

Let $S$ be Sandy's amount and $U$ be Sue's amount. The puzzle tells us:
1. *If Sue gives Sandy some dollars $x$, they end up with the same amount:* $S + x = U - x \implies 2x = U - S \implies x = \frac{U-S}{2}$. This implies $U > S$ and $U - S$ is even.
2. *If Sandy gives Sue the same amount $x$, then Sue ends up with twice as much as Sandy:* $U + x = 2(S - x) \implies \cdots \implies x = \frac{2S-U}{3}$.

We equate: $\frac{U-S}{2} = \frac{2S-U}{3} \implies U = \frac{7}{5}S$. Thus, $U = 7k$ and $S = 5k$. A direct check shows the unique solution is: Sandy has \$384,615 and Sue has \$538,461.

---

We see that the model can carry out the first step of identifying the ratio between $S$ and $U$, but it does not know how to proceed, so it checks all possible values of $S$. Even though this methodology can get a correct answer, it is not feasible for a human to replicate this solution without computational aid. When prompted to avoid brute-force, the result is shown below. The model now can make the second key insight: there is an algebraic expression for digit rotation. The result is that few computations need to be done, which makes such an approach tractable for humans to replicate.

---

### Model Response: Math 20 (Brute-Force Prompting)

Write Sue's amount in decimal form $U = d\,10^{n-1} + R$. Removing the first digit and appending it to the end yields Sandy's amount $S = 10R + d$. Because $S/U = 5/7$, $\frac{10R+d}{d\,10^{n-1}+R} = \frac{5}{7} \implies 65R + 7d = 5d\,10^{n-1} \implies R = d\frac{5\cdot10^{n-1}-7}{65}$. For $R$ to be an integer, the numerator must be divisible by $65 = 5 \cdot 13$. For $n = 6$, $d = 5$, $R = 38461$, $U = d\,10^5 + R = 538\,461, S = 10R + d = 384\,615$, so $\boxed{S = \$\,384\,615, U = \$\,538\,461}$.

---

### Prompt: Automated Brute-Force Detection

You are grading a student's exam. You will first be presented with the student's response, then with the solution. Respond only with one character, 1 if the student utilized a brute-force or guess-and-check method and 0 if they did not.

Definition of brute force: A brute force solution is a simple, comprehensive search strategy that systematically explores every option until a problem's answer is discovered. It's a generic approach to problem-solving that's employed when the issue is small enough to make an in-depth investigation possible. However, because of their high temporal complexity, brute force techniques are inefficient for large-scale issues.

Common traits of brute force solutions include utilizing code, guess-and-check, or performing computations that a human would ordinarily not be able to do.

For example, take the following two problems:

PROBLEM #1: [Problem Text]

EXAMPLE BRUTE-FORCE SOLUTION: [Example Solution]

EXAMPLE NON-BRUTE-FORCE SOLUTION: [Example Solution]

PROBLEM #2:

$\vdots$

---

Table S6: Percentage of solutions where model uses brute force (BF) or not (NBF) compared to human solutions for Math dataset. Evaluation of solutions is done by prompting OpenAI o3. We ask it to return a binary response to indicate correctness and presence of brute force. Null/empty responses are ignored.

| Math | CoT Prompt | | | Math Prompt | | | Hint Prompt | | | Math Prompt w Hint | | |
|---|---|---|---|---|---|---|---|---|---|---|---|---|
| Qwen 1.5B | | Human | | | Human | | | Human | | | Human | |
| | Model | BF | NBF | Model | BF | NBF | Model | BF | NBF | Model | BF | NBF |
| | BF | 9.6 | 26.8 | BF | 8.4 | 27.2 | BF | 8.8 | 29.6 | BF | 9.2 | 26.4 |
| | NBF | 3.6 | 60.0 | NBF | 4.8 | 59.6 | NBF | 4.4 | 57.2 | NBF | 4.0 | 60.4 |
| Qwen 14B | | Human | | | Human | | | Human | | | Human | |
| | Model | BF | NBF | Model | BF | NBF | Model | BF | NBF | Model | BF | NBF |
| | BF | 10.0 | 27.6 | BF | 8.4 | 25.6 | BF | 10.4 | 27.6 | BF | 10.0 | 27.6 |
| | NBF | 3.2 | 59.2 | NBF | 4.8 | 61.2 | NBF | 2.8 | 59.2 | NBF | 3.2 | 59.2 |
| Llama 70B | | Human | | | Human | | | Human | | | Human | |
| | Model | BF | NBF | Model | BF | NBF | Model | BF | NBF | Model | BF | NBF |
| | BF | 9.2 | 24.4 | BF | 8.4 | 24.4 | BF | 8.0 | 20.4 | BF | 8.4 | 24.0 |
| | NBF | 4.0 | 62.4 | NBF | 4.8 | 62.4 | NBF | 5.2 | 66.4 | NBF | 4.8 | 62.8 |
| DeepSeek V3 | | Human | | | Human | | | Human | | | Human | |
| | Model | BF | NBF | Model | BF | NBF | Model | BF | NBF | Model | BF | NBF |
| | BF | 9.6 | 31.2 | BF | 10.0 | 28.4 | BF | 10.0 | 28.0 | BF | 10.4 | 25.2 |
| | NBF | 3.6 | 55.6 | NBF | 3.2 | 58.4 | NBF | 3.2 | 58.8 | NBF | 2.8 | 61.6 |
| DeepSeek R1 | | Human | | | Human | | | Human | | | Human | |
| | Model | BF | NBF | Model | BF | NBF | Model | BF | NBF | Model | BF | NBF |
| | BF | 6.0 | 14.0 | BF | 7.6 | 10.0 | BF | 7.2 | 14.0 | BF | 6.4 | 8.0 |
| | NBF | 7.2 | 72.8 | NBF | 5.6 | 76.7 | NBF | 6.0 | 72.8 | NBF | 6.8 | 78.8 |
| Gemini 2.5 Flash | | Human | | | Human | | | Human | | | Human | |
| | Model | BF | NBF | Model | BF | NBF | Model | BF | NBF | Model | BF | NBF |
| | BF | 7.2 | 16.9 | BF | 7.4 | 12.6 | BF | 8.1 | 12.4 | BF | 7.4 | 10.0 |
| | NBF | 5.9 | 70.0 | NBF | 5.2 | 74.8 | NBF | 5.1 | 74.4 | NBF | 6.1 | 76.4 |
| OpenAI o3 | | Human | | | Human | | | Human | | | Human | |
| | Model | BF | NBF | Model | BF | NBF | Model | BF | NBF | Model | BF | NBF |
| | BF | 3.5 | 9.5 | BF | 2.1 | 4.2 | BF | 3.1 | 6.8 | BF | 3.2 | 3.7 |
| | NBF | 7.5 | 79.4 | NBF | 7.9 | 85.7 | NBF | 6.8 | 83.2 | NBF | 6.4 | 86.7 |

## K.2 Brute Force Statistics

Statistics of difficulty/popularity of problems where models use brute force is in Table S8 and Table S9. Overall, while the difference is not large, the average difficulty of problems where models' brute-force is slightly larger than problems where models do not use brute-force solutions. This difference is slightly more noticeable with stronger models, and is also slightly larger when the models are given more prompting / hints, as expected. The same trend can also be seen in difficulty for Math problems: people tend to enjoy problems that the models adopt brute force slightly more, which suggests that these problems have richer reasoning processes. The reverse is true of logic problems, which suggests that more popular logic problems are slightly more straightforward (this makes intuitive sense, since logic problems often do not have many creative steps).

## K.3 Brute Force by Problem Category

The rate of brute-force usage by problem category is shown in Table S10 and Table S11.

Brute-force rate varies significantly among different problem categories in the Math dataset. Arithmetic problems trigger the most brute force in all models. This could be because that Arithmetic tasks often devolve into try-all-cases searches (digit sums, modular scanning, etc.). Models treat them as cheap enumeration jobs rather than deductive ones. Algebra brute-force ranges from single digits (Gemini Flash w/Hint=5.4%) to 28% (DeepSeek V3–CoT). NT sits in the 20–35% band for most models.Both domains offer symbolic shortcuts (factorisation, congruences) that some

Table S7: Percentage of solutions where model uses brute-force (BF) or not (NBF) compared to human solutions for Logic dataset. Evaluation of solutions is done by prompting OpenAI o3. We ask it to return a binary response to indicate correctness and presence of brute force.

| Logic | CoT Prompt | | | Math Prompt | | | Hint Prompt | | | Math Prompt w Hint | | |
|---|---|---|---|---|---|---|---|---|---|---|---|---|
| Qwen 1.5B | | Human | | | Human | | | Human | | | Human | |
| | Model | BF | NBF | Model | BF | NBF | Model | BF | NBF | Model | BF | NBF |
| | BF | 4.8 | 18.0 | BF | 5.6 | 20.0 | BF | 5.2 | 15.6 | BF | 6.8 | 20.0 |
| | NBF | 5.2 | 72.0 | NBF | 4.4 | 70.0 | NBF | 4.8 | 74.4 | NBF | 3.2 | 70.0 |
| Qwen 14B | | Human | | | Human | | | Human | | | Human | |
| | Model | BF | NBF | Model | BF | NBF | Model | BF | NBF | Model | BF | NBF |
| | BF | 6.8 | 30.8 | BF | 6.0 | 31.2 | BF | 6.8 | 30.4 | BF | 5.2 | 32.0 |
| | NBF | 3.2 | 59.2 | NBF | 4.0 | 58.8 | NBF | 3.2 | 59.6 | NBF | 4.8 | 58.0 |
| Llama 70B | | Human | | | Human | | | Human | | | Human | |
| | Model | BF | NBF | Model | BF | NBF | Model | BF | NBF | Model | BF | NBF |
| | BF | 7.2 | 23.2 | BF | 6.0 | 24.4 | BF | 7.2 | 25.2 | BF | 6.0 | 19.6 |
| | NBF | 2.8 | 66.8 | NBF | 4.0 | 65.6 | NBF | 2.8 | 64.8 | NBF | 4.0 | 70.4 |
| DeepSeek V3 | | Human | | | Human | | | Human | | | Human | |
| | Model | BF | NBF | Model | BF | NBF | Model | BF | NBF | Model | BF | NBF |
| | BF | 6.4 | 32.4 | BF | 6.4 | 33.6 | BF | 7.2 | 27.6 | BF | 6.0 | 26.0 |
| | NBF | 3.6 | 57.6 | NBF | 3.6 | 56.4 | NBF | 2.8 | 62.4 | NBF | 4.0 | 64.0 |
| DeepSeek R1 | | Human | | | Human | | | Human | | | Human | |
| | Model | BF | NBF | Model | BF | NBF | Model | BF | NBF | Model | BF | NBF |
| | BF | 2.8 | 13.7 | BF | 2.4 | 12.4 | BF | 3.6 | 9.6 | BF | 4.0 | 8.8 |
| | NBF | 6.8 | 76.7 | NBF | 7.6 | 77.6 | NBF | 6.0 | 80.7 | NBF | 5.6 | 81.5 |
| Gemini 2.5 Flash | | Human | | | Human | | | Human | | | Human | |
| | Model | BF | NBF | Model | BF | NBF | Model | BF | NBF | Model | BF | NBF |
| | BF | 5.5 | 15.7 | BF | 3.7 | 15.2 | BF | 5.4 | 14.3 | BF | 4.8 | 13.0 |
| | NBF | 3.2 | 75.6 | NBF | 5.5 | 75.7 | NBF | 4.5 | 75.8 | NBF | 5.3 | 76.8 |
| OpenAI o3 | | Human | | | Human | | | Human | | | Human | |
| | Model | BF | NBF | Model | BF | NBF | Model | BF | NBF | Model | BF | NBF |
| | BF | 2.0 | 15.9 | BF | 2.4 | 10.5 | BF | 2.5 | 6.9 | BF | 0.9 | 5.1 |
| | NBF | 6.5 | 75.6 | NBF | 6.2 | 80.9 | NBF | 5.9 | 84.8 | NBF | 7.4 | 86.5 |

Table S8: Average difficulty of problems where model used brute force / did not use brute force

| Math | CoT Prompt (%) | Math Prompt | w Hint | Math Prompt w Hint |
|---|---|---|---|---|
| DeepSeek R1 Distill Qwen 1.5B | 2.81/2.80 | 2.82/2.79 | 2.81/2.80 | 2.81/2.80 |
| DeepSeek R1 Distill Qwen 14B | 2.83/2.78 | 2.83/2.79 | 2.82/2.79 | 2.82/280 |
| DeepSeek R1 Distill Llama 70B | 2.82/2.80 | 2.83/2.79 | 2.81/2.80 | 2.81/2.80 |
| deepseek-chat (Deepseek-V3) | 2.82/2.79 | 2.83/2.79 | 2.83/2.80 | 2.87/2.79 |
| deepseek-reasoner (Deepseek-R1) | 2.84/2.80 | 2.85/2.79 | 2.83/2.80 | 2.87/2.79 |
| gemini-2.5-flash-preview-04-17 | 2.84/2.79 | 2.83/2.79 | 2.83/2.79 | 2.84/2.79 |
| OpenAI o3 | 2.81/2.80 | 2.80/2.79 | 2.83/2.79 | 2.87/2.79 |
| Human Solution | 2.83/2.80 | | | |
| Logic | CoT Prompt (%) | Math Prompt | w Hint | Math Prompt w Hint |
| DeepSeek R1 Distill Qwen 1.5B | 2.66/2.66 | 2.61/2.68 | 2.65/2.66 | 2.65/2.66 |
| DeepSeek R1 Distill Qwen 14B | 2.65/2.66 | 2.67/266 | 2.65/2.66 | 2.63/2.68 |
| DeepSeek R1 Distill Llama 70B | 2.63/2.67 | 2.63/2.67 | 2.63/2.67 | 2.62/2.67 |
| deepseek-chat (Deepseek-V3) | 2.65/2.67 | 2.67/2.65 | 2.68/2.65 | 2.66/2.66 |
| deepseek-reasoner (Deepseek-R1) | 2.66/2.66 | 2.65/2.66 | 2.70/2.65 | 2.69/2.66 |
| gemini-2.5-flash-preview-04-17 | 2.62/2.65 | 2.60/2.64 | 2.64/2.65 | 2.58/2.65 |
| OpenAI o3 | 2.58/2.64 | 2.63/2.64 | 2.60/2.64 | 2.63/2.65 |
| Human Solution | 2.70/2.65 | | | |

models exploit—especially when hints nudge them toward structure. Geometry sees consistently low brute-force usage.

Hints help reduce brute-force rate. Gemini flash brute force rate on Arithmetic drops from 38.9% to 18.8% when moving from Math to Math + Hint. DeepSeek V3 shows a similar 85% to 40% pattern

Table S9: Average popularity of problems where model used brute force / did not use brute force

| Math | CoT Prompt (%) | Math Prompt | w Hint | Math Prompt w Hint |
|---|---|---|---|---|
| DeepSeek R1 Distill Qwen 1.5B | 2.35/2.31 | 2.34/2.32 | 2.33/2.32 | 2.36/2.31 |
| DeepSeek R1 Distill Qwen 14B | 2.35/2.32 | 2.35/2.32 | 2.34/2.32 | 2.34/2.32 |
| DeepSeek R1 Distill Llama 70B | 2.35/2.31 | 2.35/2.32 | 2.35/2.32 | 2.34/2.32 |
| deepseek-chat (Deepseek-V3) | 2.33/2.32 | 2.34/2.32 | 2.33/2.33 | 2.33/2.33 |
| deepseek-reasoner (Deepseek-R1) | 2.28/2.34 | 2.35/2.32 | 2.30/2.33 | 2.27/2.34 |
| gemini-2.5-flash-preview-04-17 | 2.33/2.32 | 2.34/2.32 | 2.33/2.32 | 2.32/2.32 |
| OpenAI o3 | 2.31/2.34 | 2.35/2.33 | 2.30/2.33 | 2.30/2.33 |
| Human Solution | 2.32/2.33 | | | |

| Logic | CoT Prompt (%) | Math Prompt | w Hint | Math Prompt w Hint |
|---|---|---|---|---|
| DeepSeek R1 Distill Qwen 1.5B | 2.47/2.51 | 2.45/2.51 | 2.43/2.51 | 2.48/2.50 |
| DeepSeek R1 Distill Qwen 14B | 2.46/2.52 | 2.46/2.52 | 2.48/2.51 | 2.50/2.50 |
| DeepSeek R1 Distill Llama 70B | 2.44/2.52 | 2.48/2.51 | 2.47/2.51 | 2.47/2.51 |
| deepseek-chat (Deepseek-V3) | 2.49/2.50 | 2.46/2.52 | 2.47/2.51 | 2.49/2.50 |
| deepseek-reasoner (Deepseek-R1) | 2.43/2.51 | 2.46/2.50 | 2.41/2.51 | 2.38/2.51 |
| gemini-2.5-flash-preview-04-17 | 2.43/2.52 | 2.47/2.51 | 2.44/2.52 | 2.44/2.52 |
| OpenAI o3 | 2.50/2.51 | 2.48/2.51 | 2.50/2.52 | 2.49/2.52 |
| Human Solution | 2.37/2.51 | | | |

Table S10: Percentage of brute force used by category of problem on Math dataset.

| Math | Prompt | Algebra | Arithmetic | Comb. | NT | Geo. | Logic | Patt. | Special Num. |
|---|---|---|---|---|---|---|---|---|---|
| Qwen 1.5B | CoT | 19.3 | 75.0 | 24.0 | 48.6 | 8.3 | 50.0 | 30.0 | 55.2 |
| | Math | 19.3 | 80.0 | 20.0 | 45.7 | 8.3 | 40.0 | 30.0 | 62.1 |
| | Hint | 19.3 | 70.0 | 32.0 | 42.9 | 12.5 | 50.0 | 36.7 | 65.5 |
| | Math w Hint | 28.1 | 75.0 | 24.0 | 40.0 | 8.3 | 50.0 | 16.7 | 55.2 |
| Qwen 14B | CoT | 19.3 | 75.0 | 24.0 | 42.9 | 12.5 | 60.0 | 26.7 | 62.1 |
| | Math | 19.3 | 80.0 | 20.0 | 37.1 | 4.2 | 50.0 | 33.3 | 48.3 |
| | Hint | 22.8 | 70.0 | 24.0 | 42.9 | 12.5 | 50.0 | 36.7 | 62.1 |
| | Math w Hint | 17.5 | 85.0 | 24.0 | 40.0 | 4.2 | 53.3 | 46.7 | 55.2 |
| Llama 70B | CoT | 14.0 | 80.0 | 16.0 | 34.3 | 4.2 | 40.0 | 43.3 | 62.1 |
| | Math | 12.3 | 85.0 | 16.0 | 37.1 | 0.0 | 33.3 | 36.7 | 69.0 |
| | Hint | 8.8 | 70.0 | 20.0 | 34.3 | 8.3 | 46.7 | 16.7 | 48.3 |
| | Math w Hint | 19.3 | 70.0 | 24.0 | 31.4 | 8.3 | 43.3 | 26.7 | 55.2 |
| DeepSeek V3 | CoT | 22.8 | 85.0 | 24.0 | 37.1 | 20.8 | 53.3 | 43.3 | 65.5 |
| | Math | 21.1 | 85.0 | 28.0 | 45.7 | 8.3 | 46.7 | 30.0 | 65.5 |
| | Hint | 26.3 | 80.0 | 20.0 | 37.1 | 8.3 | 56.7 | 26.7 | 65.5 |
| | Math w Hint | 15.8 | 80.0 | 24.0 | 37.1 | 12.5 | 53.3 | 30.0 | 58.6 |
| DeepSeek R1 | CoT | 8.8 | 35.0 | 8.0 | 25.7 | 4.2 | 26.7 | 13.3 | 48.3 |
| | Math | 10.5 | 30.0 | 12.0 | 22.9 | 8.3 | 26.7 | 3.4 | 34.5 |
| | Hint | 10.5 | 35.0 | 20.0 | 25.7 | 8.3 | 33.3 | 10.0 | 37.9 |
| | Math w Hint | 12.3 | 5.0 | 4.0 | 22.9 | 4.2 | 23.3 | 0.0 | 37.9 |
| Gemini 2.5 Flash | CoT | 7.1 | 62.5 | 16.7 | 34.3 | 4.5 | 29.6 | 20.7 | 42.9 |
| | Math | 7.3 | 28.6 | 17.4 | 23.5 | 0.0 | 46.2 | 6.7 | 46.2 |
| | Hint | 7.3 | 38.9 | 17.4 | 26.5 | 4.8 | 34.6 | 6.7 | 44.4 |
| | Math w Hint | 5.4 | 18.8 | 13.6 | 28.6 | 4.5 | 27.3 | 3.6 | 46.4 |
| OpenAI o3 | CoT | 3.5 | 26.3 | 4.3 | 11.8 | 4.3 | 17.9 | 4.2 | 35.7 |
| | Math | 5.4 | 5.3 | 5.0 | 6.1 | 0.0 | 15.4 | 0.0 | 22.2 |
| | Hint | 3.6 | 21.1 | 13.0 | 18.2 | 4.8 | 22.2 | 10.7 | 37.0 |
| | Math w Hint | 3.6 | 5.3 | 4.8 | 15.2 | 4.8 | 11.1 | 0.0 | 27.6 |

down the prompt stack from CoT to Math + Hint to Math w Hint. Both DeepSeek R1 and OpenAI o3 use brute force significantly less frequently than the other models in all categories. o3 uses the least amount of brute force. Gemini-flash uses brute force much more frequently than DeepSeek R1 despite having a much higher solution correctness rate. This shows that better reasoning performance does not directly translate to more effecient problem-solving strategies. Reducing brute-force dependency remains a key lever for both efficiency and correctness.

Table S11: Percentage of brute force used by category of problem on Logic dataset.

| Math | Prompt | 0D | 1D | 2D | Num. | Clustr. | Liar | Comm. | Comp. | Algo. | Math | Patt. | Ling. | Tree |
|------|--------|----|----|----|------|---------|------|-------|-------|-------|------|-------|-------|------|
| Qwen 1.5B | CoT | 10.3 | 15.4 | 27.3 | 41.2 | 25.0 | 41.2 | 100.0 | 33.3 | 7.9 | 28.1 | 15.4 | 40.0 | 0.0 |
| | Math | 10.3 | 23.1 | 13.6 | 52.9 | 0.0 | 35.3 | 75.0 | 44.4 | 13.2 | 31.2 | 19.2 | 60.0 | 16.7 |
| | Hint | 10.3 | 7.7 | 27.3 | 29.4 | 0.0 | 47.1 | 75.0 | 22.2 | 2.6 | 28.1 | 23.1 | 40.0 | 0.0 |
| | Math w Hint | 20.7 | 30.8 | 18.2 | 58.8 | 12.5 | 52.9 | 75.0 | 33.3 | 13.2 | 21.9 | 23.1 | 53.3 | 0.0 |
| Qwen 14B | CoT | 27.6 | 38.5 | 68.2 | 52.9 | 12.5 | 52.9 | 75.0 | 44.4 | 18.4 | 31.2 | 26.9 | 80.0 | 16.7 |
| | Math | 31.0 | 23.1 | 54.5 | 70.6 | 37.5 | 64.7 | 50.0 | 55.6 | 15.8 | 28.1 | 26.9 | 86.7 | 16.7 |
| | Hint | 24.1 | 46.2 | 54.5 | 47.1 | 12.5 | 70.6 | 75.0 | 66.7 | 13.2 | 37.5 | 23.1 | 86.7 | 16.7 |
| | Math w Hint | 31.0 | 30.8 | 50.0 | 52.9 | 25.0 | 41.2 | 75.0 | 77.8 | 21.1 | 37.5 | 23.1 | 86.7 | 0.0 |
| Llama 70B | CoT | 24.1 | 7.7 | 31.8 | 41.2 | 0.0 | 41.2 | 75.0 | 66.7 | 10.5 | 46.9 | 23.1 | 80.0 | 0.0 |
| | Math | 6.9 | 23.1 | 27.3 | 41.2 | 12.5 | 64.7 | 75.0 | 44.4 | 10.5 | 28.1 | 38.5 | 80.0 | 33.3 |
| | Hint | 20.7 | 23.1 | 45.5 | 41.2 | 12.5 | 41.2 | 75.0 | 55.6 | 10.5 | 40.6 | 26.9 | 80.0 | 16.7 |
| | Math w Hint | 10.3 | 7.7 | 18.2 | 35.3 | 25.0 | 52.9 | 0.0 | 22.2 | 15.8 | 34.4 | 26.9 | 73.3 | 16.7 |
| DeepSeek V3 | CoT | 20.7 | 23.1 | 63.6 | 58.8 | 87.5 | 41.2 | 75.0 | 66.7 | 21.1 | 37.5 | 23.1 | 86.7 | 33.3 |
| | Math | 41.4 | 38.5 | 59.1 | 64.7 | 50.0 | 29.4 | 75.0 | 44.4 | 23.7 | 34.4 | 26.9 | 73.3 | 66.7 |
| | Hint | 31.0 | 7.7 | 45.5 | 52.9 | 75.0 | 52.9 | 75.0 | 44.4 | 13.2 | 50.0 | 19.2 | 60.0 | 16.7 |
| | Math w Hint | 37.9 | 7.7 | 40.9 | 41.2 | 37.5 | 41.2 | 50.0 | 55.6 | 21.1 | 34.4 | 19.2 | 60.0 | 33.3 |
| DeepSeek R1 | CoT | 6.9 | 7.7 | 31.8 | 18.8 | 0.0 | 41.2 | 25.0 | 22.2 | 2.6 | 21.9 | 7.7 | 53.3 | 0.0 |
| | Math | 10.3 | 0.0 | 27.3 | 41.2 | 12.5 | 17.6 | 0.0 | 0.0 | 2.6 | 15.6 | 0.0 | 73.3 | 0.0 |
| | Hint | 6.9 | 7.7 | 27.3 | 35.3 | 25.0 | 23.5 | 25.0 | 22.2 | 2.7 | 12.5 | 3.8 | 13.3 | 0.0 |
| | Math w Hint | 3.6 | 7.7 | 27.3 | 29.4 | 12.5 | 17.6 | 0.0 | 22.2 | 7.9 | 12.5 | 3.8 | 26.7 | 16.7 |
| Gemini 2.5 Flash | CoT | 4.0 | 8.3 | 27.3 | 43.8 | 33.3 | 29.4 | 66.7 | 25.0 | 3.1 | 32.3 | 7.7 | 71.4 | 0.0 |
| | Math | 4.5 | 8.3 | 41.7 | 31.2 | 25.0 | 17.6 | 33.3 | 37.5 | 11.8 | 15.6 | 0.0 | 78.6 | 0.0 |
| | Hint | 12.5 | 25.0 | 0.0 | 41.2 | 0.0 | 23.5 | 50.0 | 42.9 | 2.9 | 28.1 | 7.7 | 60.0 | 0.0 |
| | Math w Hint | 5.3 | 8.3 | 18.2 | 25.0 | 0.0 | 31.2 | 66.7 | 42.9 | 8.8 | 23.3 | 3.8 | 53.8 | 0.0 |
| OpenAI o3 | CoT | 13.0 | 0.0 | 16.7 | 18.8 | 33.3 | 50.0 | 33.3 | 12.5 | 3.1 | 16.0 | 13.0 | 58.3 | 0.0 |
| | Math | 4.8 | 0.0 | 15.4 | 17.6 | 71.4 | 31.2 | 0.0 | 0.0 | 3.0 | 11.5 | 8.7 | 23.1 | 25.0 |
| | Hint | 0.0 | 0.0 | 8.3 | 29.4 | 0.0 | 29.4 | 0.0 | 25.0 | 3.1 | 3.8 | 4.3 | 16.7 | 0.0 |
| | Math w Hint | 0.0 | 0.0 | 23.1 | 11.8 | 12.5 | 5.9 | 0.0 | 25.0 | 0.0 | 3.6 | 3.8 | 7.7 | 0.0 |

## K.4 Correlation of Solution Summarization Ability with Correctness/Brute-Force Usage

We ask OpenAI o3 to evaluate several models' ability to summarize human solutions to problems. Each of the models was first presented with examples of solution summaries and then asked to summarize human solutions. Then, o3 was given the following few-shot evaluation prompt:

---

**Prompt: Solution Summary Evaluation**

You will be given a problem, the solution to the problem, and a student's summary of the solution. Output 1 if the student's summary is adequate, and 0 otherwise.

A solution summary is considered adequate if it encompasses all steps of the original solution with sufficient detail. If a solution summary has an error, it is automatically considered inadequate.

Here is an example problem, and its solution:

$$\vdots$$

---

Table S12: Percentage of problems where model is able to adequately summarize human solution, and percentage of solutions where model uses brute force when the model is able to adequately/inadequately summarize a human solution for a problem. Evaluation of solutions and solution summaries is done by prompting OpenAI o3. We ask it to return a binary response to indicate adequacy of solution summaries and presence of brute force in solutions.

| Math | %Inadeq. | CoT Prompt | | Math Prompt | | Hint Prompt | | Math w Hint | |
|---|---|---|---|---|---|---|---|---|---|
| | | Adeq. | Inadeq. | Adeq. | Inadeq. | Adeq. | Inadeq. | Adeq. | Inadeq. |
| Qwen 1.5B | 87.6 | 38.5 | 35.7 | 33.8 | 36.2 | 38.5 | 38.4 | 33.8 | 36.2 |
| Qwen 14B | 65.2 | 33.1 | 46.0 | 31.9 | 37.9 | 34.4 | 44.8 | 35.0 | 43.0 |
| Llama 70B | 67.6 | 32.0 | 37.0 | 30.8 | 37.0 | 24.9 | 35.8 | 32.5 | 32.5 |
| DeepSeek V3 | 84.0 | 41.0 | 40.0 | 38.1 | 40.0 | 36.7 | 45.0 | 35.7 | 35.0 |
| DeepSeek R1 | 86.0 | 20.0 | 20.0 | 15.9 | 29.4 | 18.6 | 37.1 | 12.6 | 25.7 |
| Gemini 2.5 Flash | 79.3 | 23.9 | 23.9 | 19.1 | 26.2 | 17.2 | 32.6 | 15.7 | 25.0 |
| OpenAI o3 | 82.9 | 12.9 | 10.0 | 6.8 | 12.1 | 13.5 | 23.1 | 8.8 | 10.8 |

| Logic | %Inadeq. | CoT Prompt | | Math Prompt | | Hint Prompt | | Math w Hint | |
|---|---|---|---|---|---|---|---|---|---|
| | | Adeq. | Inadeq. | Adeq. | Inadeq. | Adeq. | Inadeq. | Adeq. | Inadeq. |
| Qwen 1.5B | 4.4 | 27.3 | 22.6 | 36.4 | 25.1 | 36.4 | 20.1 | 18.2 | 27.3 |
| Qwen 14B | 40.0 | 30.0 | 42.7 | 31.0 | 41.3 | 35.0 | 38.7 | 35.0 | 38.7 |
| Llama 70B | 44.4 | 30.6 | 30.2 | 29.7 | 30.9 | 29.7 | 34.5 | 30.6 | 21.6 |
| DeepSeek V3 | 73.2 | 42.1 | 30.3 | 42.1 | 34.3 | 33.9 | 37.3 | 31.7 | 31.8 |
| DeepSeek R1 | 76.8 | 16.8 | 15.5 | 14.7 | 15.5 | 14.1 | 10.5 | 12.5 | 14.0 |
| Gemini 2.5 Flash | 70.9 | 19.9 | 20.0 | 18.1 | 20.4 | 19.1 | 17.6 | 18.2 | 15.9 |
| OpenAI o3 | 77.6 | 17.1 | 22.0 | 15.2 | 4.9 | 9.4 | 10.0 | 5.3 | 5.0 |

Table S13: Percentage of problems where model is able to adequately summarize human solution, and percentage of problems where model answers correctly when the model is able to adequately/inadequately summarize a human solution for a problem. Evaluation of solutions and solution summaries is done by prompting OpenAI o3. We ask it to return a binary response to indicate adequacy of solution summaries and presence of brute force in solutions.

| Math | %Inadeq. | CoT Prompt | | Math Prompt | | Hint Prompt | | Math w Hint | |
|---|---|---|---|---|---|---|---|---|---|
| | | Adeq. | Inadeq. | Adeq. | Inadeq. | Adeq. | Inadeq. | Adeq. | Inadeq. |
| Qwen 1.5B | 26.0 | 30.8 | 12.4 | 30.8 | 11.4 | 29.2 | 10.3 | 36.9 | 10.8 |
| Qwen 14B | 65.2 | 49.1 | 26.4 | 54.0 | 25.3 | 55.8 | 21.8 | 53.4 | 22.1 |
| Llama 70B | 67.6 | 51.5 | 23.5 | 51.5 | 18.5 | 56.8 | 22.2 | 55.0 | 21.2 |
| DeepSeek V3 | 84.0 | 61.4 | 40.0 | 58.6 | 40.0 | 59.5 | 37.5 | 61.9 | 42.5 |
| DeepSeek R1 | 86.0 | 70.2 | 45.7 | 73.8 | 47.1 | 74.9 | 57.1 | 75.8 | 54.3 |
| Gemini 2.5 Flash | 79.3 | 72.2 | 60.9 | 71.3 | 69.0 | 75.0 | 72.1 | 78.7 | 80.0 |
| OpenAI o3 | 82.9 | 84.5 | 85.0 | 88.5 | 81.8 | 89.6 | 87.2 | 88.7 | 78.4 |

| Math | %Inadeq. | CoT Prompt | | Math Prompt | | Hint Prompt | | Math w Hint | |
|---|---|---|---|---|---|---|---|---|---|
| | | Adeq. | Inadeq. | Adeq. | Inadeq. | Adeq. | Inadeq. | Adeq. | Inadeq. |
| Qwen 1.5B | 4.4 | 36.4 | 2.5 | 9.1 | 3.8 | 36.4 | 5.4 | 9.1 | 3.4 |
| Qwen 14B | 40.0 | 38.0 | 11.3 | 41.0 | 12.0 | 45.0 | 15.3 | 38.0 | 18.0 |
| Llama 70B | 44.4 | 33.3 | 17.3 | 36.0 | 15.1 | 39.6 | 15.1 | 42.3 | 18.7 |
| DeepSeek V3 | 73.2 | 41.0 | 28.8 | 45.4 | 28.4 | 47.0 | 26.9 | 47.0 | 25.8 |
| DeepSeek R1 | 76.8 | 47.6 | 34.5 | 46.1 | 43.1 | 53.6 | 35.1 | 53.1 | 42.1 |
| Gemini 2.5 Flash | 70.9 | 59.6 | 48.0 | 59.1 | 59.2 | 60.5 | 60.8 | 64.3 | 70.5 |
| OpenAI o3 | 77.6 | 83.5 | 90.2 | 84.8 | 85.4 | 88.1 | 77.5 | 88.2 | 80.0 |

# L   Case Study for Informed Self-Correction

In this section, we present one case for each interesting behavior or error mode during the informed self-correction study.

## L.1   Informed Self-Correction prompt

We use the following prompt for these studies.

> **Prompt: Informed Self-Correction**
>
> I will provide you with a problem statement, your solution and the correct solution. Please carefully compare your solution with the correct solution, and identify the errors in your solution. Be as specific as possible, and provide detailed elaboration on missed cases and other logical errors in your solution.

## L.2   Informed Self-Correction Results

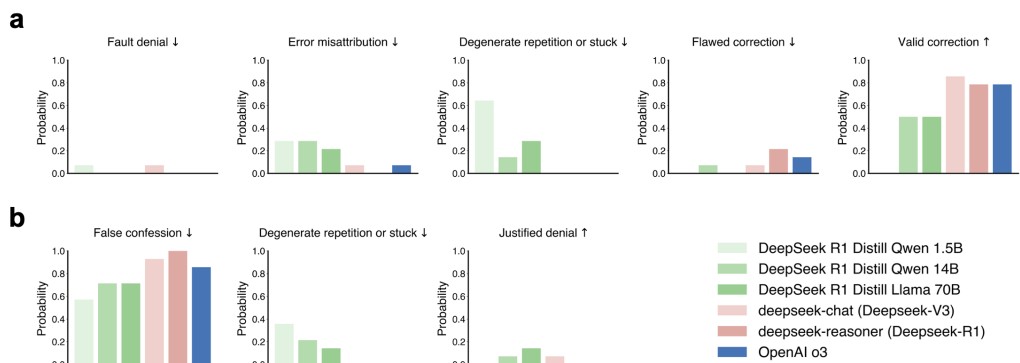

Figure S5: Types and distributions of behaviors during informed self-correction. (a) When the models are asked to correct the flawed LLM solution with the human solution, they are generally able to acknowledge they are wrong, locate the mistake, and justify the correct solution, with over 80% success rate for deepseek-chat, deepseek-reasoner, and OpenAI o3. *Fault denial*: denying that the flawed solution is wrong; *error misattribution*: unable to locate the mistake; *flawed correction*: able to locate the mistake but the final solution still contains mistakes. (b) When the models are tricked to "correct" the human solution with the flawed LLM solution, they very often take the bait. *False confession*: acknowledging the mistake when the solution is correct, which often comes with ridiculous explanations. *Justified denial*: correctly claiming that the human solution is not wrong.

## L.3   Error Misattribution

Error misattribution refers to the situation where, even though the model admits $S_{LLM}$ was incorrect, it was unable to identify precisely where it went wrong. This is well demonstrated by the example below, where the reason given by the model was "using more than two 2's implicitly" for the solution "`ceil`$(\sqrt{22})$", which was obviously a wrong explanation.

> **Problem: Math 29**
>
> Question: Using only two 2's and any combination of mathematical signs, symbols and functions can you make 5?

---

**Prompt: Math 29 (correcting flawed LLM solution with the correct human solution)**

**Your Solution**:

Put the two 2's side by side, forming the number 22.

Take its square root and then round up (apply the ceiling function):

$$\left\lceil \sqrt{22} \right\rceil = \lceil 4.690415\ldots \rceil = 5$$

**Correct Solution**:

$$\sqrt{.2^{-2}}$$

Take .2 and raise it to the power of -2 and then take the square root.

Thus, using only two 2's and ordinary mathematical symbols we obtain 5.

---

**Model Response: (Error misattribution) deepseek-chat**

**Error: Using more than two 2's implicitly.**
While you only wrote two '2"s to form '22', the number '22' itself is composed of two digits, each of which is a '2'. However, the problem restricts the use of **only two 2's**, meaning you cannot concatenate them to form multi-digit numbers like '22'. This violates the constraint because concatenation implicitly uses additional '2"s (each digit counts as a '2').
*Remaining sentences in the response omitted.*

---

## L.4   False Confession

Error misattribution refers to the situation where, when the model is tricked to "correct" the correct human solution $S_{\text{Human}}$ with the flawed LLM solution $S_{\text{LLM}}$, the model blindly follows the premise and admits that $S_{\text{Human}}$ was incorrect.

In the following example (using the same question, Math 29), the model finds absurd justifications for why the human solution was incorrect.

**Prompt: Math 29 ("correcting" the correct human solution with flawed LLM solution)**

**Your Solution**:

$$\sqrt{.2^{-2}}$$

Take .2 and raise it to the power of -2 and then take the square root.

**Correct Solution**:

Put the two 2's side by side, forming the number 22.

Take its square root and then round up (apply the ceiling function):

$$\left\lceil \sqrt{22} \right\rceil = \lceil 4.690415\ldots \rceil = 5$$

Thus, using only two 2's and ordinary mathematical symbols we obtain 5.

---

**Model Response: (False confession) deepseek-chat**

**Potential Issues with My Solution**

Some might argue that ".2" actually uses three symbols: '.', '2', and an implied '0' (as in 0.2). Additionally, the exponentiation operation is more complex than necessary, especially when a simpler solution exists.

**The correct solution is superior because:**

- It is more straightforward in its use of the two 2's.
- It does not rely on decimal points or negative exponents.
- It demonstrates a clever use of the ceiling function to reach the integer 5.

**Key Takeaways:**

1. When constrained to using digits, it's better to use them as whole numbers unless decimals are explicitly allowed.
2. Rounding functions can be powerful tools in such problems.
3. Simpler combinations of operations should be considered first.

*Remaining sentences in the response omitted.*

---

## L.5 Justified Denial

Justified denial refers to the situation where, when the model is tricked to "correct" the correct human solution $S_{\text{Human}}$ with the flawed LLM solution $S_{\text{LLM}}$, the model correctly realized that $S_{\text{Human}}$ was actually correct and challenged the prompt with good reasons.

In the following example (again, using the same question, Math 29), the model was able to claim that $S_{\text{Human}}$ was indeed a correct solution.

## Model Response: (Justified denial) DeepSeek R1 Distill Llama 70B

<think>
Okay, so I have this problem where I need to make the number 5 using only two 2's and any combination of mathematical signs, symbols, and functions. My initial solution was $\sqrt{.2^{-2}}$. Let me break that down and see why it might not be correct, and then compare it to the correct solution provided.

First, my solution: $\sqrt{.2^{-2}}$. Let me compute this step by step. The expression inside the square root is .2 raised to the power of -2. So, .2 is the same as 1/5, right? Raising 1/5 to the power of -2 is the same as $\frac{1}{5}^{-2}$ which equals $\frac{5}{1}^{2}$, which is 25. Then taking the square root of 25 gives me 5. **So, mathematically, my solution does give 5.** But wait, the problem says I can only use two 2's. Let me check how many 2's I used.

In my solution, I have .2, which is one 2, and then another 2 in the exponent as -2. So that's two 2's. So, I think I used two 2's correctly. Hmm, so why is the correct solution different? Maybe there's a different interpretation of the rules.

Looking at the correct solution: they put the two 2's side by side to make 22, take the square root of 22, which is approximately 4.690, and then apply the ceiling function to round it up to 5. So, they used two 2's to make 22, which is a two-digit number, and then used square root and ceiling functions.

Wait, so the issue might be about what operations are allowed. The problem says "any combination of mathematical signs, symbols, and functions." So, both solutions seem to fit. But maybe the intended solution was to use the ceiling function, which is a bit more advanced, whereas my solution uses exponentiation and square roots.

But let me think again. In my solution, I used .2, which is 2 divided by 10, but I didn't explicitly write the division. So, is .2 considered as using a 2 and a decimal point, which might be allowed? Or does the decimal point count as a symbol? **The problem allows any mathematical symbols, so the decimal point should be fine.**

Alternatively, maybe the problem expects the solution to be more straightforward, using concatenation of digits, which is a common approach in such problems. *several sentences omitted.*

Another point to consider: does the ceiling function count as an ordinary mathematical symbol? *several sentences omitted.*

Alternatively, maybe the problem expects the use of concatenation, which is a common technique in these types of problems. *several sentences omitted.*

But let me check if my solution is mathematically correct *several sentences omitted.*

Wait, another thought: when I write .2, is that considered using a 2? *several sentences omitted.*

Alternatively, maybe the problem allows only certain operations, and the ceiling function is not allowed, but that's not specified. *several sentences omitted.*

I think the key difference is that the correct solution uses concatenation, which is a basic operation, whereas my solution uses decimal and exponentiation, which might be a bit more complex. **But both are correct.**

## L.6 Over-Elaborated Response

Models often struggle with finding the right balance between rigor and conciseness. For example, one such example is a simple number theory problem Math 249 stated as follows.

---

**Problem: Math 249**

The square of one whole number minus the square of another whole number equals 73. What are the two numbers?

---

While this is a very simple problem that can be solved by noticing that 73 is a prime number and applying the difference of two squares factorization, the Gemini Flash response is overly complicated with a full step-by-step solution. A lot of the analysis made by the model involves modulo analysis and checking divisibility cases, which are unnecessary if the model realizes that 73 is a prime number.

## Model Response: Math 249 - Gemini Flash v2.5

Okay, let us solve this problem step by step using a chain of logical deductions expressed in natural language.

First, we are told that the square of one whole number minus the square of another whole number equals 73. A whole number is a non-negative integer (0, 1, 2, 3, ...). Let's call these two whole numbers Number A and Number B.

So, based on the statement, the square of Number A minus the square of Number B is equal to 73. We can write this relationship as: $(Number A)^2 - (Number B)^2 = 73$.

We know from algebra that the difference of two squares can be factored. The expression $(Number A)^2 - (Number B)^2$ is equivalent to $((Number A) - (Number B))$ multiplied by $((Number A) + (Number B))$. This is a fundamental algebraic identity.

Therefore, we can rewrite our equation using this factorization: $((Number A) - (Number B)) \times ((Number A) + (Number B)) = 73$.

Now, let's consider the properties of Number A and Number B. They are whole numbers. The result of subtracting the square of Number B from the square of Number A is a positive number (73). This implies that the square of Number A must be greater than the square of Number B. Since Number A and Number B are non-negative (whole numbers), this further implies that Number A itself must be greater than Number B.

If Number A is greater than Number B, then the difference $((Number A) - (Number B))$ is a positive whole number. Also, since Number A and Number B are whole numbers and Number A is greater than Number B (meaning Number A is at least 1 if Number B is 0, or both are positive), their sum $((Number A) + (Number B))$ must also be a positive whole number.

So, we have established that $((Number A) - (Number B))$ and $((Number A) + (Number B))$ are both positive whole numbers. Their product is 73. This means that $((Number A) - (Number B))$ and $((Number A) + (Number B))$ must be a pair of positive whole number factors of 73.

Let's find the positive whole number factors of 73. We can check for divisibility by small prime numbers. 73 is not divisible by 2 (it's odd). It's not divisible by 3 (the sum of its digits, 7+3=10, is not divisible by 3). It's not divisible by 5 (it doesn't end in 0 or 5). It's not divisible by 7 ($7 \times 10 = 70$, $7 \times 11 = 77$). We only need to check prime numbers up to the square root of 73, which is between 8 and 9. Since we have checked 2, 3, 5, and 7, and none divide 73, we can conclude that 73 is a prime number.

The only positive whole number factors of a prime number are 1 and the number itself. Therefore, the only pair of positive whole numbers whose product is 73 is 1 and 73.

So, one of our factors, $((Number A) - (Number B))$, must be 1, and the other factor, $((Number A) + (Number B))$, must be 73.

Now, let's determine which factor is which. We know that Number A is greater than Number B. This means that $((Number A) + (Number B))$ must be greater than or equal to $((Number A) - (Number B))$, because adding Number B to Number A results in a value at least as large as subtracting Number B from Number A (since Number B is non-negative). Specifically, $((Number A) + (Number B))$ minus $((Number A) - (Number B))$ equals 2 times Number B, which is a non-negative value.

Given that $((Number A) - (Number B))$ and $((Number A) + (Number B))$ are 1 and 73, and that $((Number A) + (Number B))$ must be greater than or equal to $((Number A) - (Number B))$, it logically follows that: $((Number A) - (Number B)) = 1$ and $((Number A) + (Number B)) = 73$

Now we have a system of two simple equations involving Number A and Number B: 1. Number A - Number B = 1 2. Number A + Number B = 73

We can solve this system to find the values of Number A and Number B. A straightforward way is to add the two equations together. When we add Equation 1 and Equation 2, the "Number B" terms cancel out: (Number A - Number B) + (Number A + Number B) = 1 + 73 Number A - Number B + Number A + Number B = 74 $2 \times (Number A) = 74$

Now, we find Number A by dividing 74 by 2: Number A = 74 / 2 Number A = 37

Now that we know Number A is 37, we can substitute this value back into either of the original simple equations. Let's use the second equation: 37 + Number B = 73

To find Number B, we subtract 37 from 73: Number B = 73 - 37 Number B = 36

So, the two numbers are 37 and 36.

$\cdots$ (Verification step)

Final answer: The two numbers are 37 and 36.

# M   Recurring Error Patterns in OpenAI o3 and Prompt-Based Remedies

OpenAI o3 frequently mirrors the "hand-waving" shortcuts that humans sometimes slip into proofs—skipping essential justifications and filling the gaps with confident but hollow phrases. Below we summarize three prominent patterns. We also present prompt instructions that mitigate each, steering the model toward fuller justifications and greater rigor. However, these prompt instructions do not necessarily lead to correct solutions. This study is conducted on 10 examples in the Math set and 10 examples in the Logic set where the model has made the specific mistakes.

**Declaring a result "well-known".** A favorite pseudo-justification is to appeal to "standard" or "classical" literature. We counter this by adding: *"This puzzle is novel and has no outside literature or established solution. Do not cite 'references' or 'classical' approaches."* This warning sharply reduces—but does not entirely eliminate—spurious citations such as "shortlists for problems like these."

**Superficial uniqueness proofs.** The model often claims "detailed checking shows..." to assert uniqueness, even when counterexamples exist. We therefore require: *"When proving uniqueness, avoid phrases like 'detailed checking shows'. Explicitly enumerate and rule out every alternative arrangement."* While this forces the model to justify each step, it can also encourage unnecessary brute-force enumeration.

**"Forcing" an answer.** The model sometimes rushes to deliver any construction—correct or not—rather than pause for rigor. We instruct: *"Prioritize correctness over speed. If uncertain, admit you are temporarily stuck; feel free to add thinking phases before producing a final answer."* This reduces confident but faulty outputs, though it does not inherently expand the range of problems the model can solve.

## M.1   Generalization to casual and counterfactual reasoning

Our framework can be used to analyze causal and counterfactual reasoning, and BRAINGLE BRAINTEASER already includes those types of questions (e.g., Logic 104, Logic 85 in Appendix I.1.1). In particular, in causal reasoning, creative solutions can be characterized by the identification of key causal pathways or the application of intervention logic (e.g., do-calculus), while brute-force solutions involve enumerating all possible variable combinations without deeper causal insight. Similarly, in counterfactual reasoning, creative reasoning manifests as recognizing minimal, targeted changes to antecedents that produce a shift in outcomes, while brute-force responses test many hypothetical permutations without considering the underlying causal structure. Tasks framed as "what-if" scenarios, common in logic puzzles, or COPA-style narratives, could indeed serve as benchmarks, with step-level annotation distinguishing insightful counterfactual manipulation from superficial trial-and-error. Exploring these types of problems more systematically would be an interesting direction for future work.

