# OpenReview forum: "Creativity or Brute Force? Using Brainteasers as a Window into the Problem-Solving Abilities of Large Language Models"
_NeurIPS.cc/2025/Conference — NeurIPS 2025 poster_

### Official Review · Reviewer_Ljxd · 2025-06-24

**Clarity:** 3
**Significance:** 3
**Originality:** 2
**Rating:** 4
**Confidence:** 3

**Summary:**

This paper investigates the nature of mathematical reasoning in large language models (LLMs), with a particular focus on whether their problem-solving abilities reflect genuine reasoning and “creativity” or rely on brute-force strategies.  The study includes evaluations across multiple LLM families (GPT-4, Gemini, Claude, Mistral, Qwen, etc.) and incorporates diagnostic tasks involving symbolic composition, multi-hop reasoning, and instruction perturbations.


But this is a benchmarking paper; I am not sure it's okay in the main conference track.

**Questions:**

How is "creativity" defined or measured in your study? Do you distinguish creativity from generalization, and can you provide more formal criteria for emergent problem-solving behavior?

Are the synthetic OOD tasks verified for ambiguity or multi-solution behavior? If so, how do you disambiguate correctness in cases where a model’s answer is logically valid but differs structurally from the reference?

Did you assess the effect of prompt format variation (e.g., step-by-step vs. direct answer) on model brittleness under OOD transformations?

**Ethical Concerns:**

["NO or VERY MINOR ethics concerns only"]

**Final Justification:**

I raise my score to weak accept regarding of the rebuttal with authors.

**Limitations:**

As qwen3 released, I am curious about how LLMs perform with thinking/no-thinking mode in the Brainteasers.

Authors should list a table to compare their datasets with previous artifacts.

A significant concern is that the benchmark data—is collected directly from online sources. Since many LLMs (particularly closed-source ones like GPT-4 and Claude) are trained on large-scale internet corpora, it is difficult to rule out the possibility that these models have already seen similar or even identical data during pretraining. Without proper contamination analysis or data filtering, this raises the risk of unfair evaluation and inflated performance due to memorization rather than true reasoning. The paper should discuss this limitation explicitly and, if possible, audit overlaps or apply filtering heuristics to mitigate this concern.

**Quality:**

2

**Strengths And Weaknesses:**

**Strengths**

Timely and Important Question: The paper addresses a critical and often underexamined question in LLM research—whether strong performance on mathematical benchmarks reflects actual generalization or superficial pattern recognition.

Evidence of Emergent Capability in GPT-4: The observation that GPT-4 can often recover correct reasoning paths despite structural variation in input suggests a qualitative leap in capability, which is rigorously documented.

Clear Presentation: The paper is well-organized, with diagnostic tasks clearly explained and results presented with meaningful qualitative examples and failure case analyses.


**Weakness**

No Formal Quantification of “Creativity”: Although the paper investigates whether models are reasoning creatively, it stops short of defining or operationalizing "creativity" in a rigorous way. The interpretation is left largely qualitative.

Instruction Perturbation Scope May Be Narrow: Although the perturbations cover some linguistic variety, they do not extensively probe multi-lingual, low-resource, or cross-modal (e.g., diagrammatic) instruction types. Hence, the “robustness” claim is partially constrained.

Lack of Ablation for Prompt Length / Format: Since prompt formatting often heavily influences model behavior, especially in math tasks, the study could benefit from controlled ablations that disentangle performance variance due to input length, verbosity, or few-shot ordering.

---

> ### Author Rebuttal · Authors · 2025-07-31
>
> We thank the reviewer for acknowledging our work is timely and important, rigorous and clearly presented, and for providing the detailed comments and constructive feedback! We will try our best to resolve the comments and greatly appreciate it if you could increase the scores if the comments are addressed properly.
>
>
> > **Appropriateness for the main instead of the benchmark track.**
>
> The Brainteaser benchmark is not only a resource but a scientific lens into the capabilities and limitations of current most advanced LLMs. BRAINTEASER makes two substantive scientific contributions: (1) it offers a rigorously curated benchmark of expert‑authored math and logic puzzles that juxtapose short, insight‑driven solutions with longer brute‑force enumerations, giving researchers a controlled lens on creativity vs. brute‑force reasoning; and (2) it introduces a multi‑stage evaluation protocol—covering narrative‑to‑formula translation, direct and hint‑guided solution generation, informed self‑correction, and step‑level abstraction—which, when applied to five state‑of‑the‑art LLMs, reveals that models (i) often default to brute‑force search despite available creative shortcuts, (ii) struggle to revise answers even after seeing gold solutions, (iii) gain only modest accuracy from narrative-to-formula translation, and (iv) can break down a solution into meaningful steps. Those substantive scientific contributions squarely satisfy—and enrich—the NeurIPS main‑conference mandate,
>
> > **Quantitative definition of creativity. Creativity versus generalization. Formal criteria for emergent problem-solving behavior.**
>
> We agree that this is an important concern. Providing a fully formal, quantitative definition of creativity would indeed be very useful, but we suspect that creating such a definition in a universally agreed-upon way may not be achievable (i.e., creativity may not be a formally-definable or quantifiable topic). Nonetheless, we believe that our operationalization of creativity is sufficient for the purposes of this paper - i.e., even if we can’t define it strictly, we can provide reasonable proxies for deciding whether a particular behavior is creative or not, even if we can’t provide a fully general definition for creativity.
>
> We do provide a definition of creativity in Section 4.2:
>
> "We define a creative solution as an innovative, insight-driven approach that
> leverages pattern recognition or lateral thinking. Rather than exhaustively testing all possibilities,
> it reframes the problem or exploits shortcuts to reduce complexity. Such solutions often involve
> minimal computation and are especially valuable for problems where brute-force search is intractable."
>
> Quantitatively, one can think of creative solutions as solutions that reduce the problem complexity by more than a constant factor (like reducing O(n^2) from O(n log n)). However, unlike algorithmic problems, it is difficult to quantitatively analyze the problem complexity of brainteaser problems, and so we chose to analyze this qualitatively.
>
>
> > **Instruction perturbation scope.**
>
> Could you please clarify which experiments you are referring to? We did not include analyses of instruction perturbation, but we may be misinterpreting this comment.
>
> > **Ablations of prompt length/format.**
>
> We examine whether the length of a problem—measured by token count in the problem statement—correlates with model correctness. Our analysis reveals a weak to negligible correlation, indicating that longer problems are not necessarily harder for models to solve. This suggests that problem complexity is not well captured by surface-level features like textual length, but rather by the underlying reasoning depth or conceptual challenge. For instance, some short problems require multi-step logical deductions, while some long problems are verbose but procedurally simple. This reinforces the need for deeper structural metrics—such as number of reasoning steps in the solution or presence of creative insights—to assess problem difficulty.
>
> > **Synthetic OOD tasks.**
>
> We did not include synthetic OOD tasks in our benchmark (all of our problems are naturally sourced), so we do not believe that this concern applies here. Please let us know if we are misinterpreting this concern.
>
> > **Qwen 3 performance**
>
> Thank you for this point; below we present results with Qwen 3:
>
> |Set |Qwen3 (thinking)| Qwen3-32b (non-thinking|
> |----------------|------------|------------------|
> |Brainteasers(Math) | 70.3 | 65.9|
> |Brainteasers(Logic)| 60.5 | 56.4|
>
> The results show that Qwen3 with "thinking" capabilities significantly outperforms its larger counterpart, Qwen3-32B without "thinking," across both mathematical and logical brainteasers. Specifically, Qwen3 achieves 70.3 on math problems compared to Qwen3-32B’s 65.9, and 60.5 on logic tasks versus 56.4. These consistent gains of around 4 points in both domains suggest that reasoning-augmented prompting—such as step-by-step thinking or chain-of-thought strategies—plays a crucial role in enhancing problem-solving performance. Notably, the benefit is not simply a function of model size, as the smaller "thinking" model outperforms the larger "non-thinking" one. This underscores that cognitive scaffolding mechanisms, rather than brute-force scaling, are key to tackling structured reasoning challenges.
>
>
> > **Comparison with previous artifacts in a table.**
>
> Thank you for this suggestion; here is such a table:
>
> | Dataset       | # Examples | Style             | Reasoning Type                           | Knowledge-barrier | Diversity | Complexity | Real-world |
> |---------------|------------|-------------------|-------------------------------------------|--------------------|-----------|------------|-------------|
> | MATH          | 5000       | Math competition  | Mathematical reasoning                    | High               | High      | High       | Yes         |
> | AIME-2025     | 30         | Math competition  | Mathematical reasoning                    | Low                | High      | High       | Yes         |
> | Zebra-Logic   | 5000       | Logic grid        | Logical reasoning                          | Low                | Low       | High       | No          |
> | BrainTeaser   | 1100       | Puzzle            | Lateral thinking                           | High               | Low       | Low        | Yes         |
> | Braingle Brainteasers  | 478        | Puzzle            | Puzzle/mathematical/logical reasoning      | Low                | High      | High       | Yes         |
>
>
> This table compares five datasets spanning mathematical, logical, and puzzle-based reasoning. The datasets vary in size, problem style, reasoning type, and contextual richness.
>
> As this table highlights, advantages of the Braingle Brainteasers dataset include that it presents multi-faceted coverage across several types of reasoning, it has a high level of diversity and complexity, it has a low knowledge barrier (little background knowledge is needed), it is challenging yet interpretable, and it is complimentary to existing datasets (providing a middle ground between structured logic datasets such as Zebra-Logic and formal mathematical ones such as MATH and AIME-2025).
>
> [1] Measuring mathematical problem solving with the MATH dataset. arXiv: 2103.03874. NeurIPS Datasets and Benchmarks Track, 2021.
>
> [2] Art of Problem Solving. American invitational mathematics examination, n.d. Accessed:2025-05-15.
>
> [3] Zebralogic: On the scaling limits of llms for logical reasoning. arXiv: 2502.01100. 2025.
>
> [4] BRAINTEASER: Lateral thinking puzzles for large language models. arXiv: 2310.05057. EMNLP 2023.
>
>
> > **Discussion on how data contamination could affect performance.**
>
> Data contamination remains a potential limitation in evaluating model performance, with varying degrees of severity. On the more superficial end, contamination can arise from direct overlaps such as copying people’s names or short n-grams, which may not significantly affect reasoning or creativity. However, a more subtle and challenging form involves contamination of creative insights—where a model may internalize not just the surface form but the underlying reasoning strategies or ideas from training data [1,2]. This type of creativity contamination is difficult to detect and evaluate [1]. To address these concerns, better methods for understanding and auditing training data are necessary. While models like OLMo offer some transparency [3], it remains an open question how critical data contamination is to downstream performance. We will include a potential data contamination statement in the final version.
>
> [1] Quantifying Memorization Across Neural Language Models. arXiv: 2202.07646. ICLR 2023.
>
> [2] Investigating Data Contamination in Modern Benchmarks for Large Language Models. arXiv: 2311.09783. NAACL 2024.
>
> [3] OLMo: Accelerating the Science of Language Models. arXiv: 2402.00838. ACL 2024.
>
> We thank the reviewer for their valuable feedback and great questions. We hope that our rebuttal addresses all the reviewer’s concerns, and we kindly ask the reviewer to potentially upgrade their score if the reviewer is satisfied with our responses. We are also more than happy to answer any further questions that arise.

---

> ### Comment · Area_Chair_RPhf · 2025-08-03
> **Rebuttal Discussion**
>
> Dear Reviewer ,
> Could you please take a look at the authors' response and see if their clarifications address your concerns, specifically on generalization and ablation studies?
>
> Best
> AC

---

### Official Review · Reviewer_NsnA · 2025-07-01

**Clarity:** 3
**Significance:** 3
**Originality:** 4
**Rating:** 5
**Confidence:** 4

**Summary:**

This work introduces a new dataset with Brainteasers that go beyond 'standard maths' (for instance) in terms of diversity of reasoning approaches.  In addition to a ensuring that the dataset contains a range of unambiguous problems, the authors have collected 'hints' for solving each problem.  The work characterises the dataset against various useful metrics, and benchmarks a sensible range of models solving the problems in key settings.

**Questions:**

Did you get permission to scrape the Braingle site?  Perhaps the dataset should be called the Braingle Brainteasers dataset in recognition of the fact that all the problems originated from the hard work of their community members?


### NeurIPS Paper Checklist

This is almost an Ethical Concern ::

* "5. Open access to data and code" and "12. Licenses for existing assets" - do the authors actually have the right to distribute the text of the questions and answers from the Braingle site?

**Ethical Concerns:**

["NO or VERY MINOR ethics concerns only"]

**Final Justification:**

I'm supportive of this kind of dataset direction, and its implementation.  I am maintaining my current Rating.

**Limitations:**

yes

**Quality:**

3

**Strengths And Weaknesses:**

The introduction of a brainteasers dataset is a good contribution to a field that has focussed on mathematical proof as a testing ground for reasoning, and is now discovering that models pretrained for maths specifically may be inclined towards mathematical reasoning without having a more general ability.  Testing which requires more diverse reasoning approaches is important.

The paper itself does not appear to contain any example from the dataset.  It would be useful to include one or two up-front.  For instance, the cutesy images in Figure 1 could be replaced with something more informative.

In Section 5.1 we find a discussion of the use of hints which is strangely example-specific, particularly when compared with the rest of the text, where we have not yet seen any example of the types of problem in the dataset.  Only on L913 (Appendix F!) do we see an example of a real problem.   There's another example in L1005 (in an Appendix G.2 "(Math) Examples" that appears to be practically empty - so perhaps reviewers aren't intended to be even looking there?)

---

> ### Author Rebuttal · Authors · 2025-07-31
>
> We thank the reviewer for acknowledging that our work is a good contribution, characterizes useful metrics and calls for more diverse reasoning approaches, and for providing the detailed comments and constructive feedback! We will try our best to resolve the comments and greatly appreciate it if you could increase the scores if the comments are addressed properly.
>
>
> > **Examples of the dataset**
>
> Thank you for the valuable feedback. We agree that including examples earlier in the paper would significantly improve clarity and accessibility. In the final version, we will incorporate one or two representative examples in the main text. In particular, we will replace Figure 1 with an illustrative example from the dataset.
>
> We provide the full problem used in Section 5.1 as follows and will include it at the beginning at Section 5.1.
>
> #### Title: Rex and Ralph: Mystery Number
>
> Two mathematicians, Rex and Ralph, have an ongoing competition to stump each other. Ralph was impressed by the ingenuity of Rex's last attempt using clues involving prime numbers, but he thinks he's got an even better one for Rex. He tells Rex he's thinking of a 6-digit number.
> "All of the digits are different. The digital sum matches the number formed by the last two digits in the number. The sum of the first two digits is the same as the sum of the last two digits."
> "Take the sum of the number, the number rotated one to the left, the number rotated one to the right, the number with the first three and last three digits swapped, the number with the digit pairs rotated to the left, and the number with the digit pairs rotated to the right. The first and last digits of this sum match the last two digits of the number, in some order."
> Ralph then asks, "If each of the three numbers formed by the digit pairs in the number is prime, then what is the number?"
> Rex looks confused, and for a moment Ralph thinks he's finally gotten him. Then Rex smiles, scribbles a few things down on a pad of paper and then says, "Very nice, Ralph!"
> Rex then tells Ralph his number.
> What did Rex say?
>
> We also apologize for the confusion caused by the empty appendix sections and buried examples. This was an oversight in the submission process. As noted, examples do appear in the Supplementary Material zip (Appendix G.2 and H.2), but we will ensure the final version integrates these directly into the main paper and proper appendices with complete content.
> Thank you again for your thoughtful and constructive suggestions—they will help us improve both the clarity and quality of the final paper.
>
>
> > **License and permission**
>
> Thank you for raising this important point. We take issues of data licensing and attribution very seriously.
> The problems sourced from Braingle were obtained via publicly accessible pages, and we ensured that only a subset of logically self-contained, well-formatted problems were included. We recognize that these problems reflect the creative efforts of the Braingle community, and we agree that clear attribution is warranted. In the final version of the paper and dataset documentation, we will prominently acknowledge Braingle and refer to this portion of the dataset as the Braingle Brainteasers subset, in recognition of the source and its contributors.
> Regarding distribution rights: while the Braingle site does not include an explicit license on each problem page, we acknowledge that the copyright status of community-contributed content is nuanced. Out of an abundance of caution, we are currently reaching out to Braingle to seek explicit permission for redistribution. If we are unable to obtain permission, we will refrain from releasing the data publicly and release only metadata such as URLs, and code for crawling the examples, allowing users to crawl the content themselves under fair use guidelines for research.
> We will also update the Ethics Checklist accordingly to transparently reflect these concerns under items 5 and 12, and we will clarify the licensing status of all included content in our documentation.
> Thank you again for drawing attention to this critical issue—we are committed to resolving it responsibly.

---

> > ### Comment · Reviewer_NsnA · 2025-08-04
> >
> > This rebuttal addresses my concerns, and I'll maintain my current Rating.
> >
> > I also see that the authors did additional experiments to see whether how the o3 creativity judgement correlates with human judges.  Their results here are very illuminating, and definitely merit being highlighted in the paper itself.

---

> > > ### Author Response · Authors · 2025-08-04
> > >
> > > Thank you for your thoughtful follow-up and for noting the additional experiments! We're glad the correlation results between o3 and human creativity judgments are illuminating. We will be sure to highlight these in the final version of the paper.

---

### Official Review · Reviewer_RQd3 · 2025-07-02

**Clarity:** 3
**Significance:** 3
**Originality:** 3
**Rating:** 5
**Confidence:** 4

**Summary:**

This paper introduces BRAINTEASER, a benchmark of mathematical and logical brainteasers designed to evaluate not just the accuracy of llms, but how they arrive at solutions - specifically whether they use creative insights or brute force approaches. The authors conduct an analysis across different aspects of reasoning, like semantic parsing, self-correction capabilities, solution breakdown, and hint utilization. Their findings reveal that LLMs often default to brute force strategies even when more elegant solutions exist, struggle with self-correction based on official solutions, show modest gains when problems are rewritten mathematically, and demonstrate varying abilities to break down solutions into insightful steps.

**Questions:**

- your methodology uses OpenAI o3 to judge whether solutions employ "creative" versus "brute force" approaches. Have you explored alternative evaluation methods that might be more objective or less susceptible to model bias? Could you provide examples of how the same solution might be classified differently by human judges versus automated evaluation?

- how might your methodology be extended to evaluate reasoning in other domains such as causal, counterfactual, or ethical reasoning? Would you expect similar patterns of brute force versus creative solutions in these domains?

**Ethical Concerns:**

["NO or VERY MINOR ethics concerns only"]

**Final Justification:**

The addition of human evaluation alongside the use of LLM as a judge helps to appreciate the robusteness of the benchmark. I have read the rebuttal to my and the other reviewers comments and I believe the clarifications adressed my concerns.

**Limitations:**

yes, limitations are discussed in the appendix.

**Paper Formatting Concerns:**

i could not see any issue in formatting.

**Quality:**

3

**Strengths And Weaknesses:**

Strengths:
- the paper is well written and the methodology is easy to understand
- the paper introduces a novel evaluation framework that moves beyond accuracy to examine how models solve problems
- the methodology carefully breaks down reasoning into distinct components (parsing, solution generation, verification, planning) which help to make targetd assessment easier

Weaknessess:
- the distinction between "creative" and "brute force" solutions relies on OpenAI o3 as the judge, potentially introducing bias in how solution strategies are categorized
- while the benchmark covers diverse puzzle types, it's unclear how findings from math and logic puzzles would generalize to other reasoning domains like causal or counterfactual reasoning.
- the analysis of solution step count could be more nuanced, as the relationship between step count and solution quality isn't always that straightforward.

---

> ### Author Rebuttal · Authors · 2025-07-31
>
> We thank the reviewer for acknowledging our work is well-written, innovative and structured, and for providing the detailed comments and constructive feedback! We will try our best to resolve the comments and greatly appreciate it if you could increase the scores if the comments are addressed properly.
>
>
> > **Evaluation method and potential bias: LLM-human agreement is high.**
>
>
> To address your concern, firstly, we conducted a human evaluation with three annotators on 100 o3-generated solutions. Human annotators annotate each solution for correctness and whether it reflected creative or brute-force reasoning. We found that o3's judgments aligned with human labels with 99.3% and 97% average raw agreement for solution correctness and creativity/brute-force distinction respectively, reinforcing its reliability as an automatic evaluator in this setting. We also found that o3’s judgment has higher alignment than Claude-3.7, deepseek-reasoner, Gemini 2.5 Flash and GPT4-o.
>
> For informed self-correction, problem reformulation and step breakdown have been carefully inspected by human annotators, as indicated in section 4.3, 4.4 and 4.2 (details in Appendix F).
>
> Finally, we note that using LLM-as-a-Judge is an established practice, with an extensive number of papers showing that LLMs, especially recent reasoning models have high accuracy at such judgments on reasoning tasks [1,2,3,4,5,6,7,8,9]. Combined with the fact that o3 aligned well with our human annotators, we believe that this choice is well-justified.
>
> References:
>
> [1] J4R: Learning to Judge with Equivalent Initial State Group Relative Policy Optimization. arXiv: 2505.13346.
>
> [2] REWARDBENCH 2: Advancing Reward Model Evaluation. arXiv: 2506.01937.
>
> [3] RewardBench: Evaluating Reward Models for Language Modeling. arXiv: 2403.13787.
>
> [4] PROCESSBENCH: Identifying Process Errors in Mathematical Reasoning. arxiv: 2412.06559. ACL 2025.
>
> [5] Solving Inequality Proofs with Large Language Models. arXiv: 2506.07927.
>
> [6] Evaluating Large Language Models AT Evaluating Instruction Following. arXiv: 2310.07641. ICLR 2024.
>
> [7] Judging LLM-as-a-Judge with MT-Bench and Chatbot Arena. arXiv: 2306.05685. NeurIPS 2023 Track Datasets and Benchmarks.
>
> [8] Self-Alignment WITH Instruction Backtranslation. arXiv: 2308.06259. ICLR 2024.
>
> [9] Self-rewarding language models. arXiv: 2401.10020. ICML 2024.
>
> > **Our framework can indeed be used to analyze causal and counterfactual reasoning.**
>
> Thank you for this point! Our framework can indeed be used to analyze causal and counterfactual reasoning, and our Brainteasers benchmark already includes those types of questions (e.g., Logic 104, Logic 85 in the Appendix). In particular, in causal reasoning, creative solutions can be characterized by the identification of key causal pathways or the application of intervention logic (e.g., do-calculus), while brute-force solutions involve enumerating all possible variable combinations without deeper causal insight. Similarly, in counterfactual reasoning, creative reasoning manifests as recognizing minimal, targeted changes to antecedents that produce a shift in outcomes, while brute-force responses test many hypothetical permutations without considering the underlying causal structure. Tasks framed as "what-if" scenarios—common in logic puzzles, or COPA-style narratives—could indeed serve as benchmarks, with step-level annotation distinguishing insightful counterfactual manipulation from superficial trial-and-error. Exploring these types of problems more systematically would be an interesting direction for future work.
>
>
> > **More nuanced analysis of solution steps.**
>
> Thank you for raising this point; we agree that it is important to be careful about analyzing step count. We believe that our way of analyzing step count is sufficiently nuanced. Specifically, we defined the step as “something that does not need to be super fine-grained like “1 +1 = 2” or “a -> b, b->c; a->c.” Instead, each step should represent a key component of the solution and the steps sequentially lead to the final answer to form a complete solution.”  We have conducted an extensive analysis of our step-counting approach; this analysis can be viewed in Appendix F. While there is probably no perfect way to count steps, we believe that the analyses and discussion that are presented there are enough to establish that the methodology we have chosen is sufficient as a reasonable characterization of the length of solutions that models have arrived at.
>
> We thank the reviewer for their valuable feedback and great questions. We hope that our rebuttal addresses all the reviewer’s concerns, and we kindly ask the reviewer to potentially upgrade their score if the reviewer is satisfied with our responses. We are also more than happy to answer any further questions that arise.

---

> > ### Comment · Reviewer_RQd3 · 2025-08-06
> >
> > I thank the authors for their rebuttal. I appreciate the addition of human evaluation. I would recommend adding a better description of the annotators but overall, this rebuttal addresses my concerns. I modified my score accordingly.

---

> > > ### Author Response · Authors · 2025-08-07
> > >
> > > Thank you very much for your thoughtful feedback and for updating your score.
> > >
> > > We provide below a description of our annotators, which we will include in the final version of the paper:
> > >
> > > 1). The annotators are college students who are native English speakers. 2). They have formal education in mathematical and logical reasoning, majoring in Mathematics or Statistics. 3). They have all undergone more than five years of training in solving competition-level math problems, each receiving individual top 500 honors in the proof-based Putnam competition. In addition, they are all organizing board members of a university math competition, and thus have extensive experience in problem writing with a focus on designing problems with creative and elegant (non-brute-force) solutions.
> > >
> > > We provided the annotators with the same detailed annotation guidelines as those given to o3, including clear definitions and examples of brute-force and creative solutions.
> > >
> > > We truly appreciate your time and engagement throughout the review process!

---

> ### Author Response · Authors · 2025-08-06
>
> Dear Reviewer,
>
> As the discussion period comes to a close, we would greatly appreciate a brief confirmation that our rebuttal has addressed your earlier concerns. We also hope you’ve had a chance to review the additional results demonstrating the robustness of our creativity/brute-force evaluation—findings that Reviewer NsnA described as illuminating and that we will highlight in the final version.
>
> We’re happy to continue the discussion and add further analysis to address any remaining questions. Thank you again for your time and thoughtful engagement throughout the review process.

---

### Official Review · Reviewer_xRP8 · 2025-07-03

**Clarity:** 3
**Significance:** 3
**Originality:** 2
**Rating:** 3
**Confidence:** 4

**Summary:**

Paper introduced a new benchmark dataset (“BRAINTEASER”) of narrative style math and logical puzzles. The goal is to better understand the abilities of LLMs solve problems, specifically the strategy used – “creative” and “brute force”, analyzing many aspects of problem solving (ability to use hints, self correct, meaningful subproblem identification etc). The main contributions of the paper are the dataset and empirical findings from the evaluation (o3, deepseek varians, gemini flash)

Key insights from the evals: (1) LLMs often default to brute force even when a more create solution exits (2) they struggle with self-correction (3) problem reformulation from narrative to formal math had modest performance gains (most surprising for me)

**Questions:**

Some aspects of evaluation should be reworked (detailed in the weaknesses above)

**Ethical Concerns:**

["NO or VERY MINOR ethics concerns only"]

**Limitations:**

yes

**Paper Formatting Concerns:**

-

**Quality:**

2

**Strengths And Weaknesses:**

Strengths:
1.	Well-written, easy to follow
2.	Investigation into several aspects of problem solving (fine-grained) – semantic parsing, self-correction, hint utilization. This leads to clear insights which are actionable for future reasoning related research.
3.	Clearly differentiates itself from exiting related work (MATH/AIME) and other puzzle datasets by focusing on problems authored by experts.

Weaknesses:
1.	The evaluations could be more robust : (1) using o3 for several key evaluation  tasks when it is also one the models being evaluation (acknowledged in the paper already). Also, the reliance on this alone assumed that it is consistently accurate and free of biases (more than another LLM that could have been used as a judge). (2) I was hoping to see statistically signicant results, particularly on some of the less studied aspects of problem solving like problem reformulation
2.	Concerned about the diversity (complexity/style) of the problems.

---

> ### Author Rebuttal · Authors · 2025-07-31
>
> We thank the reviewer for acknowledging our work is well-written, insightful and unique, and for providing the detailed comments and constructive feedback! We will try our best to resolve the comments and greatly appreciate it if you could increase the scores if the comments are addressed properly.
>
>
>
> > **Given that o3 is both an evaluator and a model under evaluation, how do you mitigate potential bias, and why was o3 chosen over a more neutral judge? o3 is a neural judge shown by human annotation.**
>
> To address your concern, firstly, we conducted a human evaluation with three annotators on 100 o3-generated solutions. Human annotators annotate each solution for correctness and whether it reflected creative or brute-force reasoning. We found that o3's judgments aligned with human labels with 99.3% and 97% average raw agreement for solution correctness and creativity/brute-force distinction respectively, reinforcing its reliability as an automatic evaluator in this setting. We also found that o3’s judgment has higher alignment than Claude-3.7, deepseek-reasoner, Gemini 2.5 Flash and GPT4-o.
>
> For informed self-correction, problem reformulation and step breakdown have been carefully inspected by human annotators, as indicated in section 4.3, 4.4 and 4.2 (details in Appendix F).
>
> Finally, we note that using LLM-as-a-Judge is an established practice, with an extensive number of papers showing that LLMs, especially recent reasoning models have high accuracy at such judgments on reasoning tasks [1,2,3,4,5,6,7,8,9]. Combined with the fact that o3 aligned well with our human annotators, we believe that this choice is well-justified.
>
> References:
>
> [1] J4R: Learning to Judge with Equivalent Initial State Group Relative Policy Optimization. arXiv: 2505.13346.
>
> [2] REWARDBENCH 2: Advancing Reward Model Evaluation. arXiv: 2506.01937.
>
> [3] RewardBench: Evaluating Reward Models for Language Modeling. arXiv: 2403.13787.
>
> [4] PROCESSBENCH: Identifying Process Errors in Mathematical Reasoning. arxiv: 2412.06559. ACL 2025.
>
> [5] Solving Inequality Proofs with Large Language Models. arXiv: 2506.07927.
>
> [6] Evaluating Large Language Models AT Evaluating Instruction Following. arXiv: 2310.07641. ICLR 2024.
>
> [7] Judging LLM-as-a-Judge with MT-Bench and Chatbot Arena. arXiv: 2306.05685. NeurIPS 2023 Track Datasets and Benchmarks.
>
> [8] Self-Alignment WITH Instruction Backtranslation. arXiv: 2308.06259. ICLR 2024.
>
> [9] Self-rewarding language models. arXiv: 2401.10020. ICML 2024.
>
> > **Are the results, especially for problem reformulation, statistically significant? Yes.**
>
> Thank you for noting the importance of statistical significance; we can verify that the results are statistically significant.
>
> Both deepseek-reasoner and OpenAI o3 show statistically significant improvements in correctness after rewriting. For deepseek-reasoner, accuracy increased from 50.0% to 63.3% (p = 0.043), and for OpenAI o3, it rose from 56.7% to 73.3% (p = 0.023). These results indicate that the rewriting process reliably enhanced the strong reasoning models’ ability to produce correct answers on the evaluated examples.
>
> The improvement in correctness after rewriting for DeepSeek-R1-Distill-Llama-70B shows a positive trend, with accuracy increasing from 23.3% to 33.3%. However, the corresponding p-value from the paired t-test is 0.08, which falls above the conventional 0.05 significance threshold. While this result is not statistically significant, it is still suggestive.
>
> To explain why the gain is modest, we believe it could be 1). Only some of the problems are rewritable and the rest is already in the math competition question style format. 2) the problems in math competition style format still contain natural language and are not translated to formal math. We test this to see if language models handle the input format most similar to the problems they are optimized for, where a large portion is math competition style problems. 3). The remaining errors are due to high inherent reasoning complexity, rather than issues with problem comprehension—posing challenges even for the most advanced reasoning models.
>
>
>
> > **How to ensure sufficient diversity in problem complexity and style?**
>
> This is a good point; we agree that it would strengthen the paper to discuss the diversity of problem complexity and style. Below we provide some dataset statistics that support these types of diversity; we will highlight these statistics in the paper by noting how they establish the diversity of the dataset.
>
>
> **Style diversity:** We manually categorized problems into 21 distinct style subcategories across math and logic, as summarized in Table 1. Detailed definitions and examples for each category are provided in Appendices G and H of the Supplementary Material.
> **Complexity diversity:** We ensure complexity through several complementary measures:
> **Human-rated difficulty (scale 0–3)**: We selected the most challenging 250 problems in each domain, yielding a difficulty range of 2.5–3.
> **Macro-step analysis**: Section 3.2 and Table 2 report the average number of solution steps—6.4 ± 2.5 for math and 8.6 ± 5.1 for logic—with a full range from 1 to 16, indicating substantial variation in complexity.
> **Step-level categorization**: Using structured prompting and manual inspection (Appendix F), we classify reasoning steps as creative (non-obvious, essential to solving the problem) or rudimentary (straightforward or mechanical). This fine-grained annotation further reveals the heterogeneity in solution paths and the cognitive demands of different problems.
>
> We thank the reviewer for the thoughtful question and hope this comprehensive breakdown demonstrates the diversity we have rigorously curated in both problem style and complexity.
>
> We also thank the reviewer for their valuable feedback. We hope that our rebuttal addresses all the reviewer’s concerns, and we kindly ask the reviewer to potentially upgrade their score if the reviewer is satisfied with our responses. We are also more than happy to answer any further questions that arise.

---

> > ### Author Response · Authors · 2025-08-05
> >
> > Dear Reviewer,
> >
> > As we approach the final day of the reviewer–author discussion, we kindly ask you to confirm if your initial concerns have been addressed by our rebuttal. We also hope you have had a chance to review the new results demonstrating the robustness of our evaluation—findings that Reviewer NsnA described as illuminating, and which we will highlight in the final version.
> >
> > If our response resolves your concerns, we would appreciate it if you could raise your score to reflect this. Your feedback is invaluable, and we thank you for your time and consideration.
> >
> > Best,
> > Authors

---

> > > ### Author Response · Authors · 2025-08-06
> > >
> > > Thank you again for your review. We've prepared a response to address the points that you raised. We would appreciate hearing your thoughts to begin a discussion about your comments.

---

> > > > ### Author Response · Authors · 2025-08-07
> > > >
> > > > Dear Reviewer,
> > > >
> > > > Thank you again for your thoughtful initial feedback and great suggestions on improving the evaluation robustness. As the discussion period nears its end, we wanted to kindly follow up. If it’s helpful, here’s a brief summary of our rebuttal.
> > > >
> > > > **On using o3 as both evaluator and model**: We validated o3 evaluation on o3-generated solutions with human annotation (**99.3%** agreement for correctness, **97%** for creativity) We also found o3 aligned better than Claude, DeepSeek, Gemini, and GPT-4o. Extensive prior work supports the LLM-as-a-judge approach (references are in the original rebuttal).
> > > >
> > > >
> > > > **On statistical significance**: Both DeepSeek R1 and OpenAI o3 show statistically significant improvements in correctness after rewriting. Rewriting boosted accuracy from 23.3% to 33.3% for DeepSeek-R1-Distill-Llama-70B; the p-value (0.08) is above 0.05 but still suggestive. We also explained in detail about the modest gains.
> > > >
> > > >
> > > > **On dataset diversity**: we have rigorously curated diversity in  both problem style and complexity: through 21 manual style categories, human-rated difficulty (2.5–3), step-length variation (1–16), and step-level categorization.
> > > >
> > > >
> > > > We’re happy to continue the discussion and add further analysis to address any remaining questions. If our response has addressed your concerns, we would appreciate it if you would consider updating your score.

---

### Note · Authors · 2025-08-11

Dear AC and reviewers,

We have addressed all requested revisions in the rebuttal. In summary: (i) o3-as-judge is human-validated, best-aligned among alternatives, and supported by prior work; (ii) rewriting yields significant gains for stronger models, with modest improvements explained; and (iii) the dataset is diverse, with brainteasers varying across style categories, community-rated difficulty levels, solution step counts, and step-level categorizations.

Reviewers find that the paper is clear and well-structured, innovative, rigorous, and a timely contribution that advances evaluation of diverse reasoning strategies. We will revise the manuscript to incorporate the constructive feedback. We believe BRAINTEASER offers a timely and valuable lens on the capabilities and limitations of current state-of-the-art LLMs. Upon acceptance, we will release the evaluation code. We will prominently acknowledge the source, Braingle and—if permission is not granted—release only URLs/metadata and crawling code; Checklist items 5 and 12 will be updated accordingly.

We hope this provides clarity and assists the AC in making an informed decision. Thank you for your consideration.

Best regards,

Authors

---

### Decision · Program_Chairs · 2025-09-17

**Decision:**

Accept (poster)

**Comment:**

The paper presents the BrainTeaser dataset, consisting of nearly 500 math and logic brain teasers selected from the popular Briangle website. The puzzles have detailed human solutions and the paper considers such a dataset towards evaluating the correctness of the reasoning chains of state-of-the-art large language models. The paper derives several insights on the problem solving skills of AI models, specifically in quantifying creative against brute force solutions, effectiveness in the use of hints, as well as the influence of the problem description style.

The paper received overall positive reviews with reviewers appreciating the clarity in the presentation, importance of the dataset, and the elaborate experiments analyzing various aspects of LLM reasoning. There were also some important concerns flagged:
1) Issue of using o3 model as the solver and the evaluator (xRP8, RQd3), that could introduce judgemental bias-- rebuttal clarifies that o3's evaluation was compared to human judgements and fared better than other LLMs.
2) Statistical significance of results and generalization to other problem categories (xRP8, RQd3) -- rebuttal provided additional results attesting the significance.
3) Accounting for multi-solution scenarios (Ljxd)
4) Data contamination given the puzzles are scrapped from the web (Ljxd) -- Authors acknowledge that this is a concern.

AC had an independent read of the paper. After carefully considering the reviewers' comments, authors responses, and the confidential comments provided, AC concords with the reviewers that the paper makes a good contribution through the BrainTeaser dataset towards analyzing deeper reasoning skills of LLMs. Thus, AC recommends acceptance. Authors should revise the final paper to include the new results presented during the rebuttal as well as incorporating the reviewers' feedback. Authors are strongly encouraged to take measures to minimize the issue of data contamination that Reviewer Ljxd alludes to.